# Vestigial auriculomotor activity indicates the direction of auditory attention in humans

**Daniel J Strauss[1]\*, Farah I Corona-Strauss[1], Andreas Schroeer[1], Philipp Flotho[1], Ronny Hannemann[2], Steven A Hackley[3]**

[1]Systems Neuroscience and Neurotechnology Unit, Faculty of Medicine, Saarland University & School of Engineering, htw saar, Homburg/Saar, Germany; [2]Audiological Research Unit, Sivantos GmbH, Erlangen, Germany; [3]Clinical and Cognitive Neuroscience Laboratory, Department of Psychological Sciences, University of Missouri, Columbia, United States

**Abstract** Unlike dogs and cats, people do not point their ears as they focus attention on novel, salient, or task-relevant stimuli. Our species may nevertheless have retained a vestigial pinna-orienting system that has persisted as a 'neural fossil' within in the brain for about 25 million years. Consistent with this hypothesis, we demonstrate that the direction of auditory attention is reflected in sustained electrical activity of muscles within the vestigial auriculomotor system. Surface electromyograms (EMGs) were taken from muscles that either move the pinna or alter its shape. To assess reflexive, stimulus-driven attention we presented novel sounds from speakers at four different lateral locations while the participants silently read a boring text in front of them. To test voluntary, goal-directed attention we instructed participants to listen to a short story coming from one of these speakers, while ignoring a competing story from the corresponding speaker on the opposite side. In both experiments, EMG recordings showed larger activity at the ear on the side of the attended stimulus, but with slightly different patterns. Upward movement (perking) differed according to the lateral focus of attention only during voluntary orienting; rearward folding of the pinna's upper-lateral edge exhibited such differences only during reflexive orienting. The existence of a pinna-orienting system in humans, one that is experimentally accessible, offers opportunities for basic as well as applied science.

**\*For correspondence:**
daniel.strauss@uni-saarland.de

## Introduction

Watching the ears allows an equestrian to gauge their mount's shifting attention. Ear movements are not a useful cue in humans or apes because higher primates have lost the ability to orient by adjusting pinna shape and focal direction. Instead we judge a person's attention by their gaze direction. In thousands of research reports each year, though, casual observation of ocular orienting is replaced by sophisticated recording techniques. We show in the present paper that similar electrical and optical techniques allow us to extract muscular correlates of pinna-orienting in our species and even render subtle pinna-orienting movements visible. Activation of the ear muscles is directionally specific and it occurs during voluntary as well as reflexive attention.

A review of research in *Hackley, 2015* on pinna-orienting in humans identified three relevant findings scattered across the preceding 100-or-so years. The first was Wilson's oculo-auricular phenomenon (*Wilson, 1908*), in which shifting the gaze hard to one side elicits a 1 to 4 mm deflection of the lateral rim of both ears. The relevance to spatial attention is uncertain, though, with diverging results across studies, for example see *Gerstle and Wilkinson, 1929*; *Urban et al., 1993*; *O'Beirne and Patuzzi, 1999*. Additional evidence comes from a 1987 study (*Hackley et al., 1987*) of the bilateral

**eLife digest** Dogs, cats, monkeys and other animals perk their ears in the direction of sounds they are interested in. Humans and their closest ape relatives, however, appear to have lost this ability. Some humans are able to wiggle their ears, suggesting that some of the brain circuits and muscles that allow automatic ear movements towards sounds are still present. This may be a 'vestigial feature', an ability that is maintained even though it no longer serves its original purpose.

Now, Strauss et al. show that vestigial movements of muscles around the ear indicate the direction of sounds a person is paying attention to. In the experiments, human volunteers tried to read a boring text while surprising sounds like a traffic jam, a baby crying, or footsteps played. During this exercise, Strauss et al. recorded the electrical activity in the muscles of their ears to see if they moved in response to the direction the sound came from. In a second set of experiments, the same electrical recordings were made as participants listened to a podcast while a second podcast was playing from a different direction. The individuals' ears were also recorded using high resolution video.

Both sets of experiments revealed tiny involuntary movements in muscles surrounding the ear closest to the direction of a sound the person is listening to. When the participants tried to listen to one podcast and tune out another, they also made ear 'perking' movements in the direction of their preferred podcast.

The results suggest that movements of the vestigial muscles in the human ear indicate the direction of sounds a person is paying attention to. These tiny movements could be used to develop better hearing aids that sense the electrical activity in the ear muscles and amplify sounds the person is trying to focus on, while minimizing other sounds.

postauricular muscle (PAM) reflex (onset latency = 10 ms) to acoustic onset transients. Increased amplitudes were observed when subjects directed their attention to a stream of tones on the same side as the recorded muscle while ignoring a competing, contralateral stream. Comparisons across left/right stimulus, attention, and PAM combinations localized modulation to the motor limb of the reflex arc. This pattern could indicate that the muscle behind an ear is primed when attention is directed toward that side. Finally, an experiment in *Stekelenburg and van Boxtel, 2002* found that the automatic capture of attention by unexpected sounds coming from a speaker hidden to the left of the participant elicited greater activity in the left than right PAM.

Apart from the research just described, functional studies of the human auriculomotor system have been mainly limited to the PAM reflex, in the context of audiometry or affective psychophysiology. The auriculomotor system lies essentially untouched in the literature. Here we present evidence that our brains retain vestigial circuitry for orienting the pinnae during both exogenous, stimulus-driven attention to brief, novel sounds and endogenous, goal-directed attention to sustained speech. We also demonstrate a complex interplay of different auricular muscles which may be causally linked to subtle movements of the pinnae.

## Results

### Experiment 1 - Exogenous attention

To examine automatic, stimulus-driven attention we used novel sounds similar to those in the *Stekelenburg and van Boxtel, 2002* study, for example traffic jam, baby crying, footsteps. However, we presented them randomly from four different speakers (at $\pm$ 30°, $\pm$ 120°; *Figure 1*) rather than just one, while the subject read a boring essay. As we were interested in the interactive role of distinct muscles in attempting to shape and point the pinnae, we recorded EMG from posterior, anterior, superior, and transverse auricular muscles (PAM, AAM, SAM, and TAM). Visual evidence had previously been limited to still photos of Wilson's oculo-auricular phenomenon (*Wilson, 1908*), so we supplemented our EMG data with videos from four high-definition cameras, see Methods and *Video 1*, *Video 2*, and *Video 3*. To confirm that our findings would generalize to different age groups, older (62.7 $\pm$ 5.9 y) as well as younger (24.1 $\pm$ 3.1 y) adults were tested.

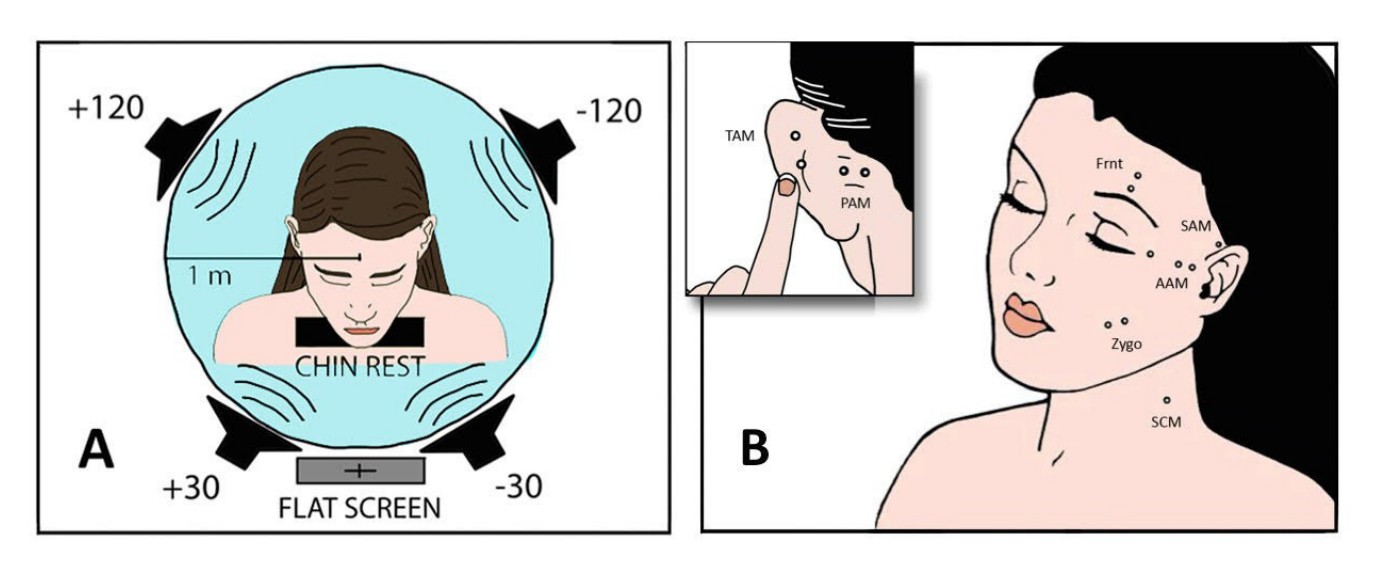

**Figure 1.** Experimental setup. (A) Four loudspeakers presented novel sounds (Exp. 1) or stories (Exp. 2) at 30° to the left or right of fixation or behind the interaural axis. Instructions, text, or fixation cross was displayed on a 55 in flat screen. (B) Surface EMGs were recorded bilaterally from four auricular muscles as well as from left zygomaticus major, frontalis, and sternocleidomastoideus, using a bandpass of 10 – 1000 Hz and a sampling rate of 9600 Hz. Separation of paired auricular electrodes was 1 cm.

The signal-averaged EMG waveforms of *Figure 2* show well-defined responses with an onset latency of about 70 ms, responses that vary in amplitude, duration, and morphology according to the relative direction of the sound source. The inter- and intra-subject variability for the analyzed auricular muscles is shown in *Figure 3* and *Figure 4*, respectively. These plots portray the consistency of the PAM, AAM, and TAM responses across stimuli and subjects, especially for stimulation from the back. For the statistical analysis, mean amplitudes were subjected to a mixed, repeated-measures analysis of variance, with factors of age group, stimulus-muscle correspondence (ipsi-/contralateral), and anterior/posterior stimulus direction. EMG amplitudes were larger for stimulus sources on the same side as the recorded ear for PAM, AAM, and TAM [$F(1, 26)$ = 47.44, 17.01, and 47.53, respectively; $p$-values < 0.001; $\eta_p^2$ = 0.65, 0.40, and 0.65] but not SAM.

Responses were also larger to sounds emanating from the back than the front speakers for PAM, AAM, and TAM [$F(1, 26)$ = 32.1, 12.0, and 19.9, respectively; $p$-values < 0.003, $\eta_p^2$ = 0.55, 0.32, and 0.43]. Posterior, ipsilateral stimulation elicited the most vigorous responses from these three muscles. In particular, the interaction between the factors ipsi/contralateral and anterior/posterior for PAM, AAM, and TAM yields $F(1, 26)$ = 40.4, 14.9, and 23.6, respectively; $p$-values < 0.002; $\eta_p^2$ = 0.61, 0.36, and 0.48. There were no interactions involving age group, but a main effect indicated that older participants had smaller AAM responses [$F(1, 26)$ = 6.0, $p$ < 0.03, $\eta_p^2$ = 0.19].

These results support the hypothesis that the human brain retains circuits that attempt to point the ears in the direction of unexpected, potentially relevant sounds. The corresponding vestigial auriculomotor drive appears to be causally linked to very small ear displacements, see *Figure 2—figure supplement 1*, *Video 1*, and *Video 2*. Having documented the existence of directionally-appropriate responses of the ear muscles to brief novel sounds, we turn now to a qualitatively distinct type of attention.

## Experiment 2 – Endogenous Attention

To examine voluntary, goal-directed attention we used the classic, dichotic-listening paradigm, see *Hillyard et al., 1973*; *Cherry, 1953*. Two competing short stories were played either over the two front speakers or the two back speakers. To increase motivation participants were allowed to choose, after a brief introduction, which of the two stories (podcasts) they would like to listen to.

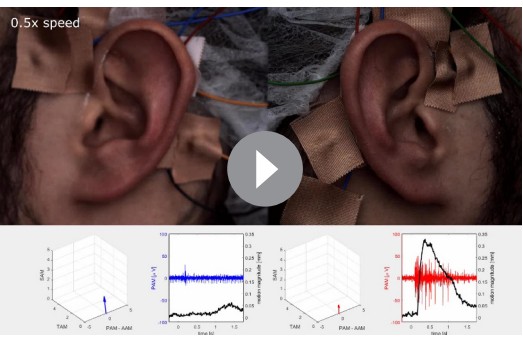

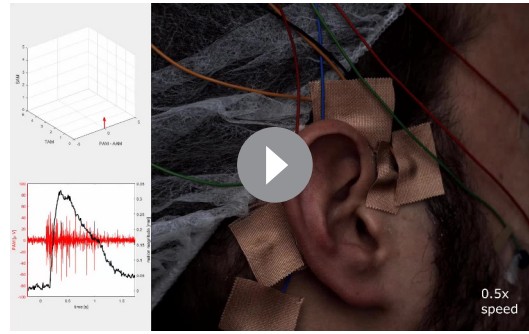

**Video 1.** Experiment 1 –Ear movement example from a trial with a novel sound at the right posterior speaker in Experiment 1. The right half of the display portrays evoked movements of the ipsilateral pinna in three ways. The large video clip of the pinna uses digital magnification to render the overall pattern of movement apparent. The color overlay in these videos indicates the motion magnitude. Just below the video and to the right, an unrectified EMG recording of the postauricular muscle is shown in co-registration with the video. The global head motion was reduced by a 2-dimensional rigid pre-registration with respect to a set of manually specified reference points on the head (see also *Figure 2—figure supplement 1*). The 3-dimensional graph medial to the 2-D graph includes a vector that indicates moment-by-moment changes in EMG activity of the superior auricular muscle (SAM, the vertical axis), transverse auricular muscle (TAM, a horizontal axis), and the difference between activity in the posterior and anterior auricular muscles (PAM-AAM, the other horizontal axis). The left half of this video gives corresponding information for the contralateral ear which, consistent with evidence presented in the main text, was not as active as the ipsilateral one.

https://elifesciences.org/articles/54536#video1

**Video 2.** Experiment 1 –The right ear example from the previous video, but with four different videos in sequence. The first video of the sequence shows the raw recording (without digital magnification). The second video shows the digitally magnified motion, the third video shows the magnified motion with color overlay as in the previous video supplement, and the fourth video shows the three dimensional motion from a different angle. This video sequence shows the impact of the digital motion magnification and the depth information about ear motion that can be derived from a stereo computer vision setup such as the one used here.

https://elifesciences.org/articles/54536#video2

They were then told which speaker that story would be presented from. Our subjects were instructed to listen carefully while looking at a fixation cross and, as in the immediately preceding study, holding their head still on a chin rest. Upon completion, a new story was picked and the listening direction was switched to one of the other speakers. Recording methods were identical to those of the exogenous experiment. Muscle activity was quantified as the mean of the absolute EMG energy over the entire course of each 5 min listening trial in *Figure 5* and for consecutive segments of 10 s duration in a temporal analysis shown in *Figure 6*.

As in the exogenous study, EMG energy at PAM and AAM was largest on the side to which attention was focused [analysis corresponding to *Figure 5*: F(1, 19) = 15.2 and 4.6, respectively; $p = 0.001$ and 0.04; $\eta_p^2 = 0.44$ and 0.20]; an effect that is particularly strong in narrow–band middle frequency components of the signal, see *Figure 5—figure supplements 1* and *2* and the tables in *Supplementary file 1*.

A different pattern emerged for the other two muscles. Whereas TAM but not SAM activity had reflected lateralization of transient, exogenous attention, the reverse was true for sustained, goal-directed attention. That is to say, mean EMG energy at SAM was larger at the ipsi– than contralateral ear [F(1, 19) = 16.3; $p = 0.001$; $\eta_p^2 = 0.46$] in Experiment 2, but there was no such difference for TAM.

Another main effect indicated that activation of all four muscles was generally enhanced when participants listened to one of the two speakers that were slightly behind as opposed to in front of them [PAM, AAM, TAM, SAM: F(1, 19) = 5.7, 3.1, 8.1, and 12.0, respectively; $p = 0.03$, 0.09, 0.01, and 0.003; $\eta_p^2 = 0.23$, 0.14, 0.30, and 0.39]. These effects did not interact with each other or with age. Although PAM activity declines over time, EMG energy of all three muscles is clearly sustained

across the 5-min sessions, see *Figure 6*. A corresponding sustained deflection of the pinna is also noticeable in the co-registered *Video 3*.

## Potential motor confounds

An alternative to the account we have been developing is that participants in Experiments 1 and 2 may have shifted their gaze toward the attended source. This would have then triggered *Wilson, 1908* phenomenon, that is auriculomotor activity secondary to large gaze shifts. To test this hypothesis, we segmented the horizontal electrooculogram (EOG) in the same way as the auricular EMG. Voltages were converted to degrees of arc separately for each participant, based on findings from a cursor tracking protocol (± 35°). *Figure 7* and *Figure 8* document a complete absence of eye movements that were systematically related to attention direction. A limitation of these findings is that electro-oculographic recordings have a resolution of only $1 - 2°$. However, gaze shifts less than 30° are rarely accompanied by auriculomotor activity, see *Urban et al., 1993*. Representative examples of macrosaccades during reading in Experiment one with co-registered auricular muscle activity can be found in *Figure 7—figure supplement 1*. There is no obvious linkage of saccades and PAM responses. Note that the mean visual angle range observed in this example generalizes across subjects, see *Figure 7—figure supplement 2*. Also when considering all the macrosaccades from all the subjects in Experiment 1, our data do not exhibit a regularity between auditory stimuli and macrosaccades, see *Figure 7—figure supplement 3*.

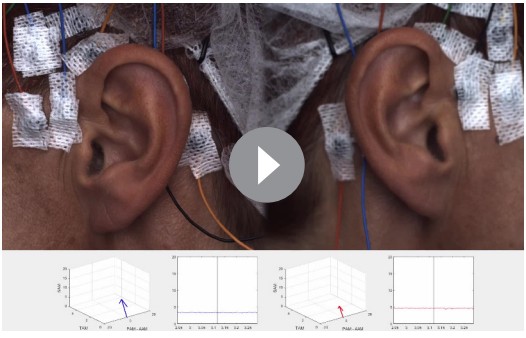

**Video 3.** Experiment 2 –Ear movement example from a participant who exhibited exceptionally large, long-lasting involuntary auricular muscle activations and ear motion during the endogenous attention task in Experiment 2. The attention of the participant was directed to the story played from the posterior right speaker. The organization of the plots and co-registration is as in *Video 1*. However, this time the raw videos without digital magnification are shown. The raw videos are played faster, time-locked to the time axis given in minutes in the one dimensional plots of the rectified postauricular muscle activity. Note that time-axis reflects the entire timeline including the instructions and the introduction to the stories before the directional listening task. The listening task started at approximately 2 min. The video also documents the end of the listening task (around 7 min) accompanied with a time–locked offset of the muscle activation and pinna displacement. A causal relation of the rectified postauricular muscle activity and the motion magnitude in the videos is clearly noticeable, especially for the ipsilateral ear.

https://elifesciences.org/articles/54536#video3

Another line of evidence that the auricular responses observed in our study were not secondary to eye movements, concerns their pattern of lateralization. Activation of TAM during Wilson's oculo-auricular phenomenon is more vigorous on the side opposite the direction of gaze, see *Gerstle and Wilkinson, 1929*; *Urban et al., 1993*. By contrast, we found in Experiment one that TAM activation was relatively enhanced at the ear on the same side as the attention-engaging sounds. The PAM component of Wilson's phenomenon does exhibit enhanced activity on the ipsilateral side, but this effect appears to be reliable only for gaze shifts greater than about 40 degrees (*Patuzzi and O'Beirne, 1999*, *Figure 4*).

Another alternative interpretation is that participants oriented not with their ears or eyes, but by lifting their chin from the chin rest and rotating their head toward the attended sound. If humans have a vestibulo-auricular response as do cats (*Tollin et al., 2009*), such head rotations could have indirectly triggered activity in the ear muscles. However, recent research has shown that azimuthal head rotations have little effect on auricular activity in humans, see *Cook and Patuzzi, 2014*. Moreover, analysis of sternocleidomastoid EMG in our data suggest that movements of the neck were rare, small, and unsystematic, see *Figure 7—figure supplements 4* and *5*. An additional statistical analysis in *Supplementary file 1* also rejects an influence of the sternocleidomastoid EMG and horizontal EOG. Finally, head rotations would have been too slow to generate the rapid responses of around 70 ms onset latency observed in the ear muscles in Experiment 1. We note with interest, though, the possibility that subtle, covert activation of head turning muscles (*Corneil et al., 2008*) might be correlated with ocular and auricular orienting. Note that there was also no corresponding

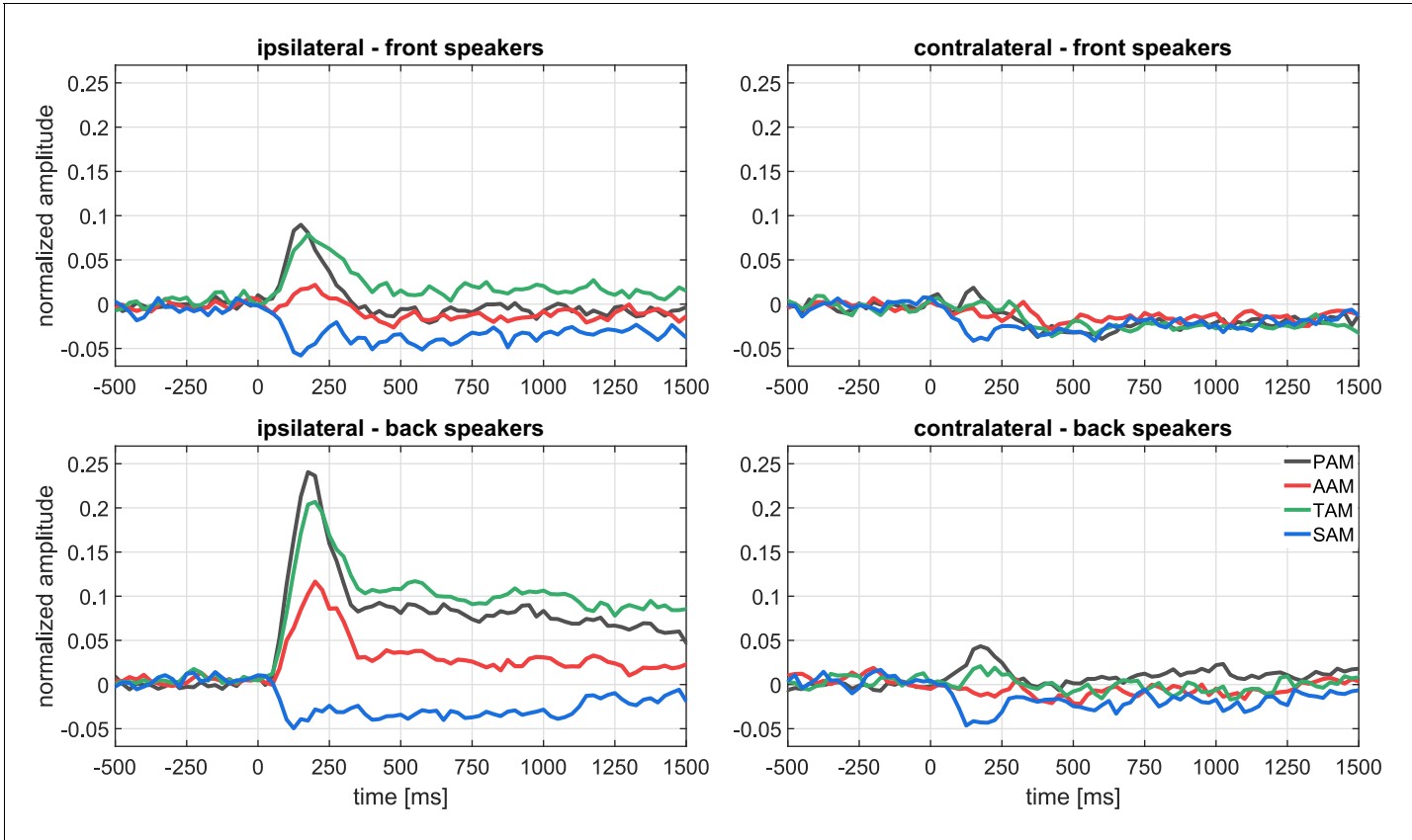

**Figure 2.** Experiment 1. Grand average (N = 28) of the baseline corrected and normalized event-related electromyograms at the four auricular muscles for the recordings ipsilateral (left panel) and contralateral (right panel) to stimulation; top: front speakers (30°), bottom: back speakers (120°). The contralateral-ipsilateral organization of our data set is justified by a preliminary analysis that obtained null effects for left–versus–right using a more complete factorial structure (left/right stimulus direction × left/right recording site). The following figure supplement is available for *Figure 2—figure supplement 1*. Analysis of video recordings from one participant who exhibited submillimeter pinna displacements in response to stimulation from the back speakers. This figure supplement is complemented by *Video 1* and *Video 2*.
The online version of this article includes the following figure supplement(s) for figure 2:

**Figure supplement 1.** Analysis of video recordings from one participant who exhibited submillimeter pinna displacements in response to stimulation in Experiment 1 (Exogenous Attention).

co–activation of the other measured (non–auricular) facial muscles, the zygomaticus and frontalis muscle, see *Figure 7—figure supplements 6–9* for Experiment one and *Figure 8—figure supplement 2* for Experiment 2.

## Discussion

These data provide compelling evidence that our brains retain, in vestigial form, circuitry for orienting the pinnae during both exogenous and endogenous modes of attention. The neural drive to our ear muscles is so weak that the actual movements (see co-registered video data in the Supplementary Information) are at least one to two orders of magnitude smaller compared to those generated during biting, smiling, grimacing, or voluntary ear-wiggling. To understand what remains of the vestigial pinna-orienting system so as to exploit it for practical or scientific purposes it is helpful to take a comparative, phylogenetic approach (*Hackley, 2015*, *Hackley et al., 2017*).

### Vestigial Pinna-Orienting

The ability to swivel and point the pinnae seems to have been lost during the transition from the primarily nocturnal lifestyles of prosimians to the diurnal ones of New World monkeys, and then, Old World monkeys (*Coleman and Ross, 2004*). Mobility continued to decline as the ears became

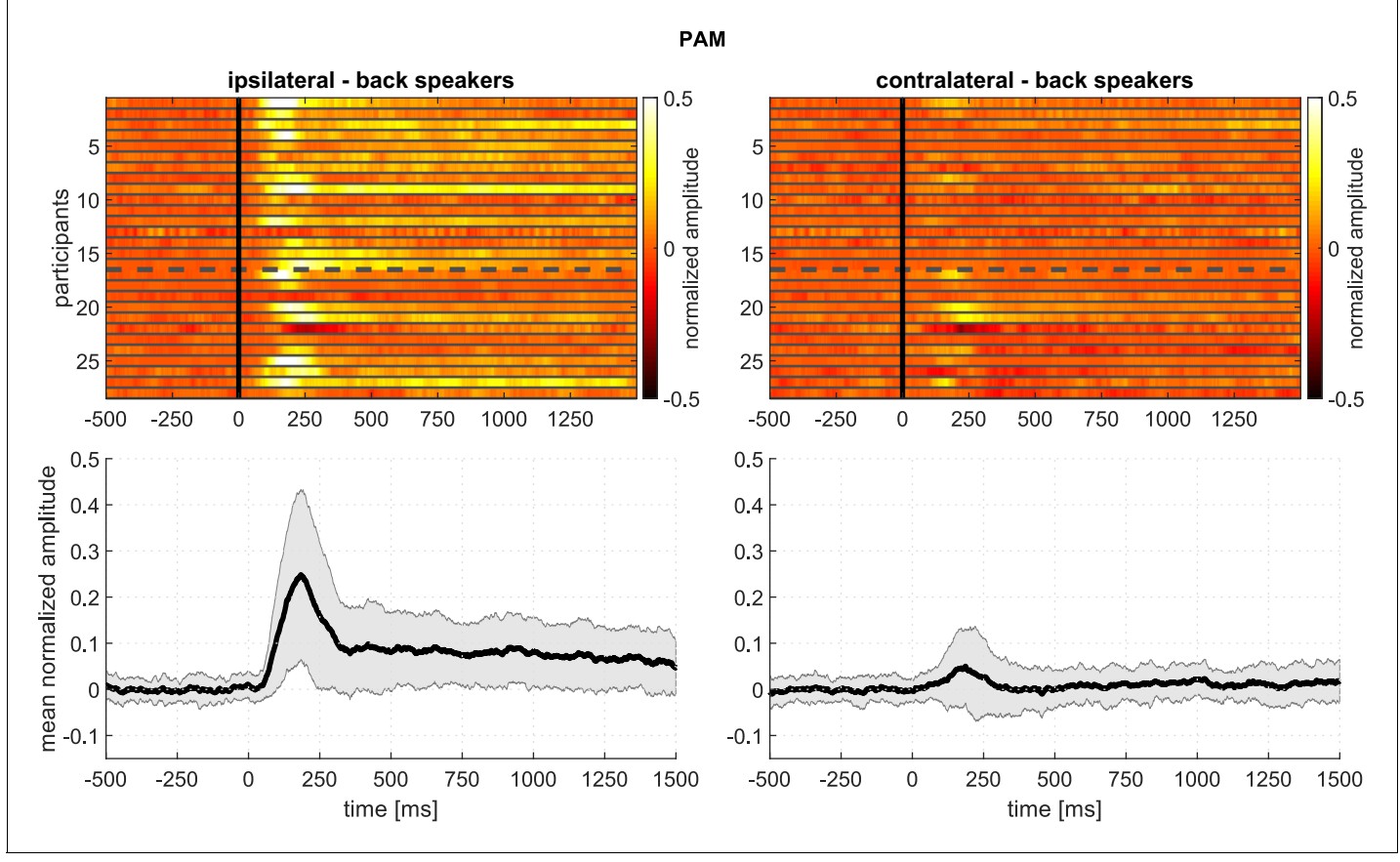

**Figure 3.** Experiment 1 – Responses of the PAM to stimuli from the back speakers, showing intersubject variability: Top panels: Every row corresponds to the averaged response of one participant. Amplitude is encoded in color. The top rows (1-16) represent younger adult participants; the bottom rows (17-28), older adults. Bottom panels: Mean and standard deviation based on the above plots. The following figure supplements are available for *Figure 3—figure supplement 1*. The described intersubject variability analysis for the AAM, *Figure 3—figure supplement 2*. SAM, and *Figure 3— figure supplement 3*. TAM.

The online version of this article includes the following figure supplement(s) for figure 3:

**Figure supplement 1.** Experiment 1 – Responses of the AAM to stimuli from the back speakers, showing intersubject variability: Top panels: Every row corresponds to the averaged response of one participant.

**Figure supplement 2.** Experiment 1 – Responses of the SAM to stimuli from the back speakers, showing intersubject variability: Top panels: Every row corresponds to the averaged response of one participant.

**Figure supplement 3.** Experiment 1 – Responses of the TAM to stimuli from the back speakers, showing intersubject variability: Top panels: Every row corresponds to the averaged response of one participant.

shorter and more rigid, see *Waller et al., 2008*; *Coleman and Ross, 2004*. The musculature degenerated. For example, an inferior auricular muscle to oppose SAM still exists in lesser apes such as gibbons and siamangs (*Burrows et al., 2011*), but not in chimpanzees (*Burrows et al., 2011*) or humans (*Cattaneo and Pavesi, 2014*). Given that head rotation has little effect on PAM activity (*Cook and Patuzzi, 2014*), it seems likely that the vestibulo–auricular reflex as documented in cats (*Tollin et al., 2009*) has not been conserved in our species. Also presumably lost is the ability to use proprioceptive information to adjust auditory processing in accordance with pinna position, orientation, and shape as documented in cats, see *Kanold and Young, 2001*. Although the ear muscles of Old World monkeys have spindles (*Lovell et al., 1977*), those of humans do not (*Cattaneo and Pavesi, 2014*).

When pinna-orienting movements became too small to modify acoustic input substantially, possibly 25 million years ago when lesser apes branched off from Old World monkeys, see *Gibbs et al., 2007* as discussed in *Hackley, 2015*, selective environmental pressure ceased. The neural system became more-or-less 'frozen' in a form optimized for controlling taller, more flexible ears, mounted

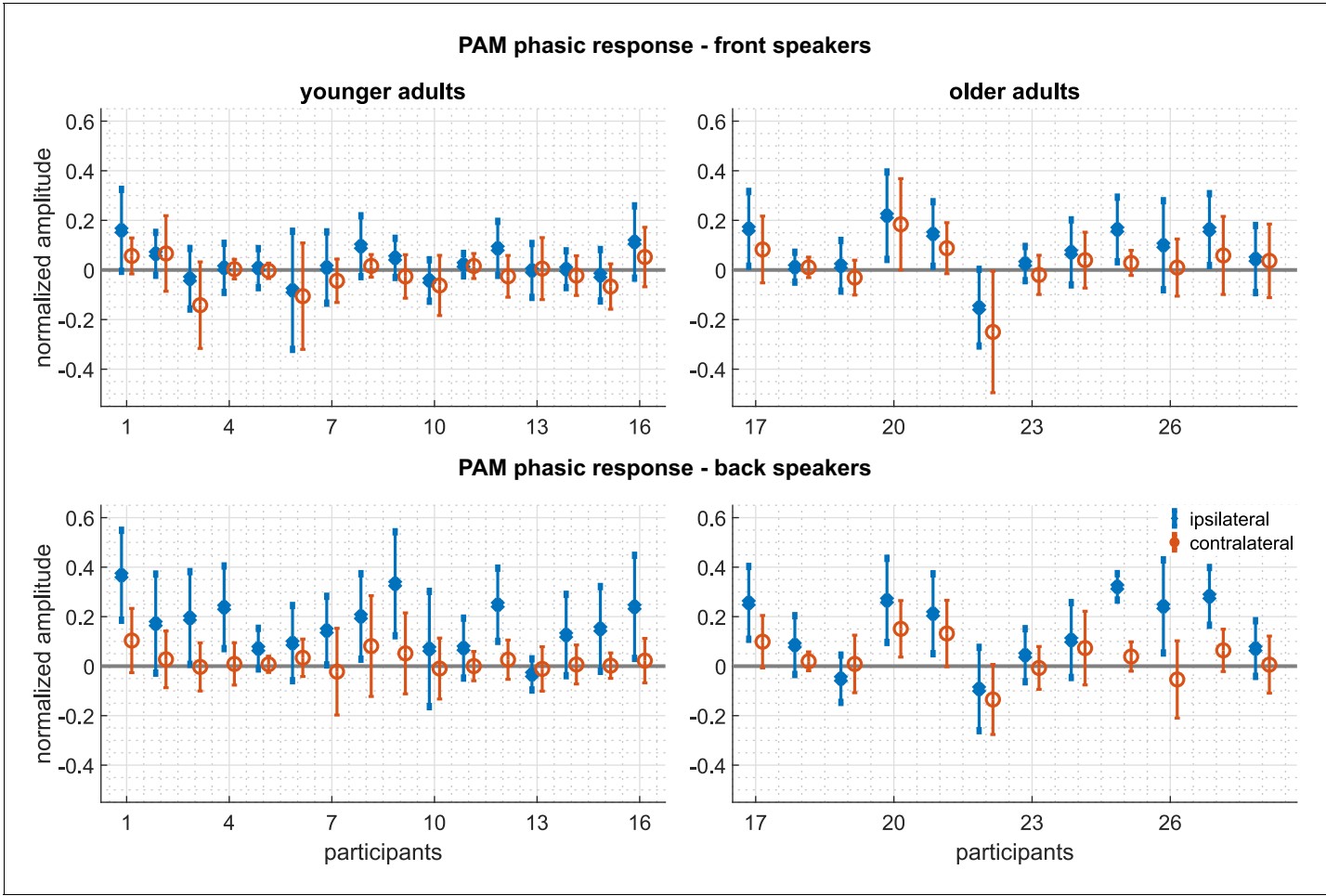

**Figure 4.** Experiment 1 – Intrasubject variability of the PAM: Mean and standard deviations of the phasic responses 50 - 300 ms) of every participant. Top panels: responses to the front speakers. Bottom panels: responses to the back speakers. Left panels: Responses of the younger adults. Right panels: responses of the older adults. Blue represents ipsilateral responses, red represents contralateral responses. The following figure supplements are available for *Figure 4—figure supplement 1*. The described intrasubject variability analysis for the AAM, *Figure 4—figure supplement 2*. SAM, and *Figure 4—figure supplement 3*. TAM.
The online version of this article includes the following figure supplement(s) for figure 4:

**Figure supplement 1.** Experiment 1 – Intrasubject variability of the AAM: Mean and standard deviations of the phasic responses ( 50 - 300 ms) of every participant.

**Figure supplement 2.** Experiment 1 – Intrasubject variability of the SAM: Mean and standard deviations of the phasic responses (50 -300 ms) of every participant.

**Figure supplement 3.** Experiment 1 – Intrasubject variability of the TAM: Mean and standard deviations of the phasic responses ( 50 - 300 ms) of every participant.

on a smaller, more spherical head. This evolutionary perspective helps us to understand the surprising finding that AAM, which pulls the base of the pinna forward, was activated in Experiment one by novel sounds coming from the rear. Co-activation of opposing muscles AAM and PAM in our remote ancestors would have reduced occlusion of the ear canal by the tragus. Note that this occlusion occurs in monkeys when contraction of the PAM homolog is unopposed, see *Waller et al., 2008*, supplementary video clip 17. In addition, PAM-AAM co-activation would have stabilized the base of the pinna and reduced myotendinous elasticity, thereby allowing quick changes in position or orientation. This perspective also illuminates our unexpected finding of ipsilateral SAM suppression in Experiment 1. A study of pinna orienting in cats, whose tall ears resemble those of prosimians, showed that they tend to tilt the ear downward slightly when orienting to a lateral target, see *Populin and Yin, 1998*.

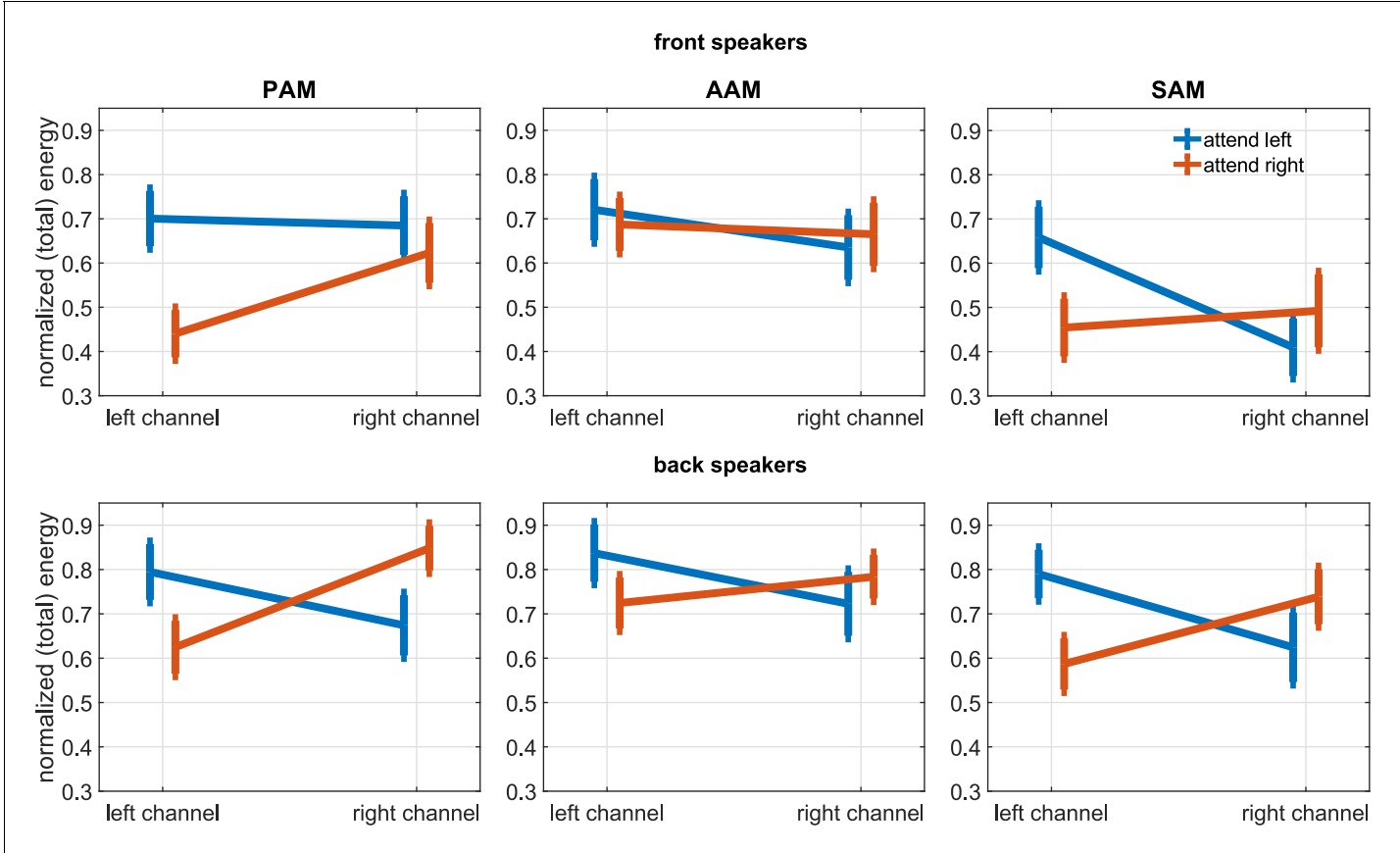

**Figure 5.** Experiment 2. Grand average of the PAM, AAM, and SAM activity when stories were played from the front (top) and back speakers (bottom). Shown is the normalized (total) energy of the left/right recording channels during attention to the left or right story (bars represent the standard error). The following figure supplements are available for *Figure 5—figure supplement 1*. Subband analysis of the described ipsi- vs. contralateral effect for PAM, AAM, and SAM; *Figure 5—figure supplement 2*. Reported results for a selected narrow frequency band.
The online version of this article includes the following figure supplement(s) for figure 5:

**Figure supplement 1.** Grand average of the PAM, AAM, and SAM activity when stories were played from front (± 30°) and back speakers (± 120°).

**Figure supplement 2.** Analogous to *Figure 5* and *Figure 6*, respectively, in the main text but for frequency band 5 (37.5 – 75 Hz): Grand average of the PAM, AAM, and SAM activity when stories were played from the back speakers (± 120°).

## Potential neural mechanisms

Neurobiologists have distinguished two types of pinna-orienting movements in cats, based on onset latency, see *Siegmund and Santibáñez, 1982*. The short-latency response is specific to auditory stimuli and is chronometrically uncorrelated with saccades toward the target. By contrast, the long-latency response can be elicited by visual as well as auditory stimuli and it is roughly synchronous with ocular orienting, see *Populin and Yin, 1998*. Using a 4-speaker set-up similar to that of the present Experiment 1, *Siegmund and Santibáñez, 1982* found cats' unconditioned pinna responses to have an EMG onset latency that averaged 78 ms, similar to our value of about 70 ms. The animals were then trained to make gaze shifts toward the sound sources. Onset latency of the auriculomotor responses dropped to a remarkable 29 ms and the responses were resistant to extinction over the course of 125 trials. Both findings were replicated by *Populin and Yin, 1998* (mean = 26 ms; failure to extinguish across 10,000 unreinforced trials). The latter authors obtained an even more rapid response (mean = 21 ms) when the sound was preceded by a visual stimulus that served as warning signal and indicated that the cat should maintain gaze at a fixation point. They argued that the short-latency pinna response is too rapid to be mediated by the brain region most centrally involved in orienting, the superior colliculus (SC). This is because an earlier study, *Populin and Yin, 1997*, had found the average first-spike latency in the relevant portion of this structure to be 19 ms.

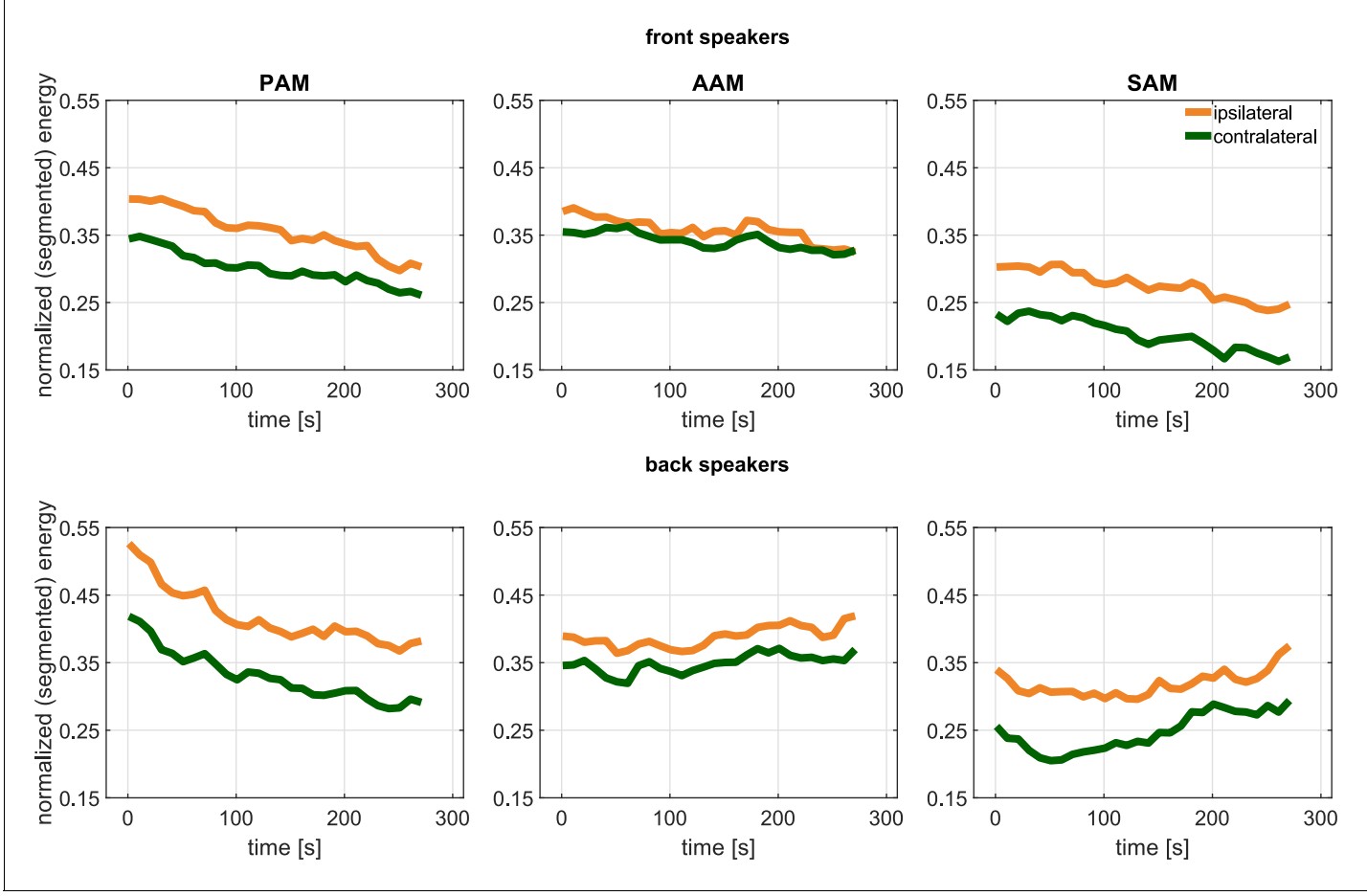

**Figure 6.** Experiment 2. Time–resolved activity (each sampling point represents the energy induced in consecutive 10 s segments) after pooling the ipsi– and contralateral signals with a segment–wise normalization for the front (top) and back speakers (bottom).

Comparisons with these cat studies suggest that our participants' auriculomotor responses may have been primarily also of the short-latency variety that is not mediated by the SC. Two of the conditions tested by *Populin and Yin, 1998* involved brief, lateralized auditory stimuli that, as in the present Experiment 1, were not task-relevant. The stimuli elicited short-latency ipsilateral pinna movements that were temporally uncorrelated with gaze shifts (see their Figures 6 and 8). During the delayed-saccade condition of their study, laterally presented sounds were task-relevant and forward fixation was required, as in our Experiment 2. Ipsilateral pinna movements triggered by onset of these sounds were of the short-latency variety (21 ms, as noted above). Subsequent, smaller movements were then observed in synchrony with ocular orienting, roughly 400 ms after the fixation point was extinguished (*Figure 7*). It is long-latency pinna movements of this sort that *Populin and Yin, 1998* suggested might be mediated by the SC.

Given the major role of the SC in controlling eye fixation (*Krauzlis et al., 2017*), this structure may also be responsible for sustained maintenance of pinna orientation, such as in Experiment 2. Pinna movements can be triggered by electrical stimulation of the deep and intermediate layers of the SC in accordance with a topographical pattern that is in register with that of eye movements (*Stein and Clamann, 1981*). Lesions of this structure reduce the likelihood of pinna orienting as well as its accuracy, see *Czihak et al., 1983*. Although monosynaptic connections from SC to auriculomotor neurons in the facial nucleus do exist (*Vidal et al., 1988*), pinna control is dominated by disynaptic pathways from the SC that include the paralemniscal, oculomotor, or pontine reticular zones, see *Henkel and Edwards, 1978*; *Takeuchi et al., 1979*; *Vidal et al., 1988*. Among these, the paralemniscal zone appears to be the most important, and its auditory input originates in the nearby nucleus sagulum, see *Henkel, 1981*.

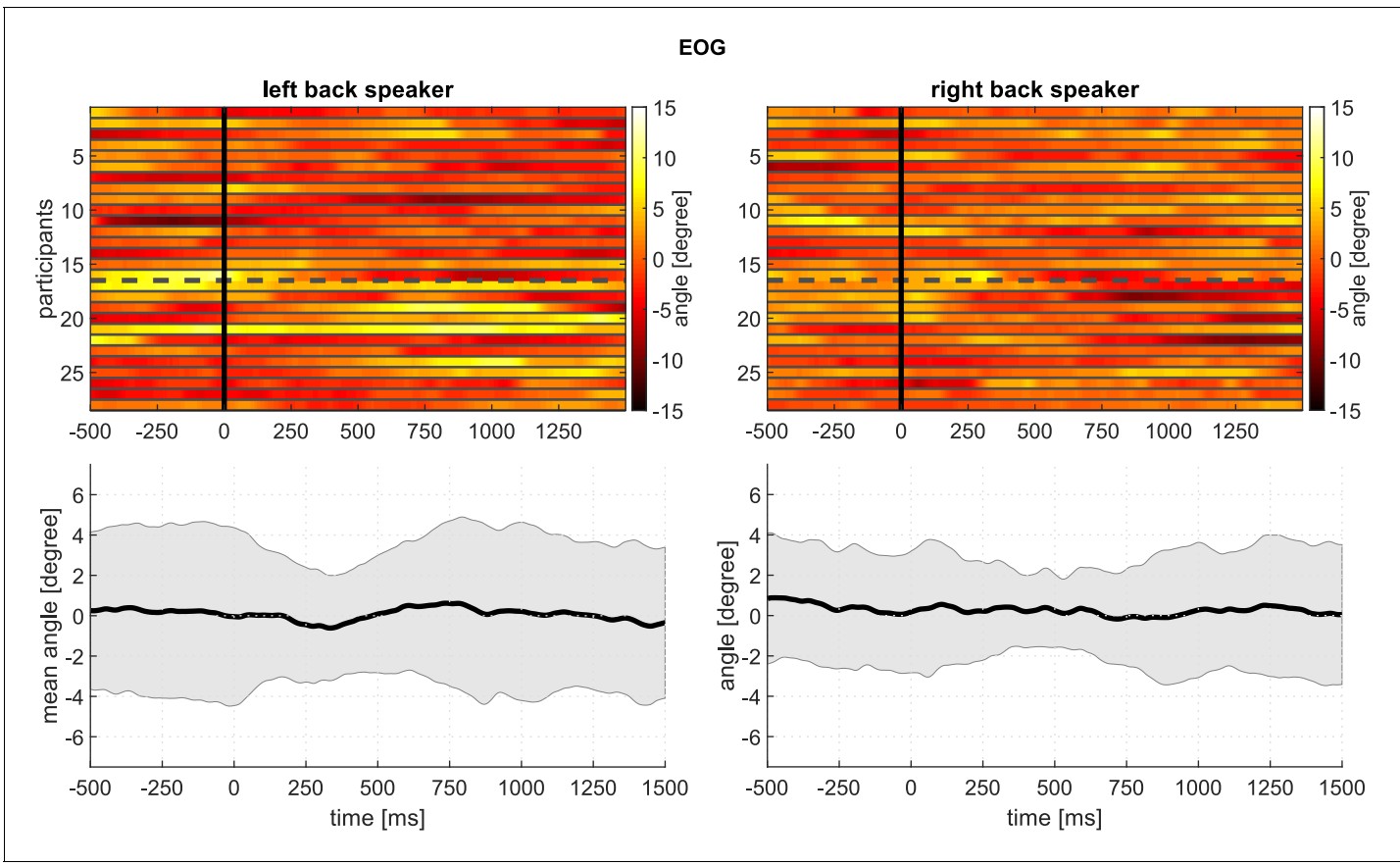

**Figure 7.** Experiment 1: Averaged horizontal EOG activity at around the time of stimulation from the back speakers, showing an apparent absence of systematic shifts in gaze direction: Top panels: Every row corresponds to the averaged response of one participant. Gaze angle is encoded in color, such that positive values (yellow) indicate rightward eye movements/positive angles. The top rows (1-16) represent younger adult participants; the bottom rows (17-28), older adults. Bottom panels: Mean and standard deviation based on the above plots.The following figure supplements are available for *Figure 7—figure supplement 1*. Macrosaccades during reading for one subject as example; *Figure 7—figure supplement 2*. Boxplots of the EOG for all subjects; *Figure 7—figure supplement 3*. Density of all detected macrosaccades during Experiment 1; *Figure 7—figure supplement 4*. Responses of the M. sternocleidomastoideus to stimuli from the front speakers; *Figure 7—figure supplement 5*. Responses of the M. sternocleidomastoideus to stimuli from the back speakers; *Figure 7—figure supplement 6*. Responses of the M. frontalis to stimuli from the front speakers; *Figure 7—figure supplement 7*. Responses of the M. frontalis to stimuli from the back speakers; *Figure 7—figure supplement 8*. Responses of the M. zygomaticus to stimuli from the front speakers; *Figure 7—figure supplement 9*. Responses of the M. zygomaticus to stimuli from the back speakers.

The online version of this article includes the following figure supplement(s) for figure 7:

**Figure supplement 1.** Macrosaccades during reading in Experiment one as recorded by means of horizontal EOG (black line) along with time-synchronized EMG from left and right PAM (blue and orange lines, respectively).

**Figure supplement 2.** Boxplots of EOG signals from all subjects in Experiment 1.

**Figure supplement 3.** Density of detected macrosaccades during Experiment 1.

**Figure supplement 4.** Experiment 1 – Responses of the M. sternocleidomastoideus (M.SCM) to stimuli from the front speakers, showing intersubject variability: Top panels: Every row corresponds to the averaged response of one participant.

**Figure supplement 5.** Experiment 1 – Responses of the M. sternocleidomastoideus (M.SCM) to stimuli from the back speakers, showing intersubject variability: Top panels: Every row corresponds to the averaged response of one participant.

**Figure supplement 6.** Experiment 1 – Responses of the frontalis muscle to stimuli from the front speakers, showing intersubject variability: Top panels: Every row corresponds to the averaged response of one participant.

**Figure supplement 7.** Experiment 1 – Responses of the frontalis muscle to stimuli from the back speakers, showing intersubject variability: Top panels: Every row corresponds to the averaged response of one participant.

**Figure supplement 8.** Experiment 1 – Responses of the zygomaticus muscle to stimuli from the front speakers, showing intersubject variability: Top panels: Every row corresponds to the averaged response of one participant.

**Figure supplement 9.** Experiment 1 – Responses of the zygomaticus muscle to stimuli from the back speakers, showing intersubject variability: Top panels: Every row corresponds to the averaged response of one participant.

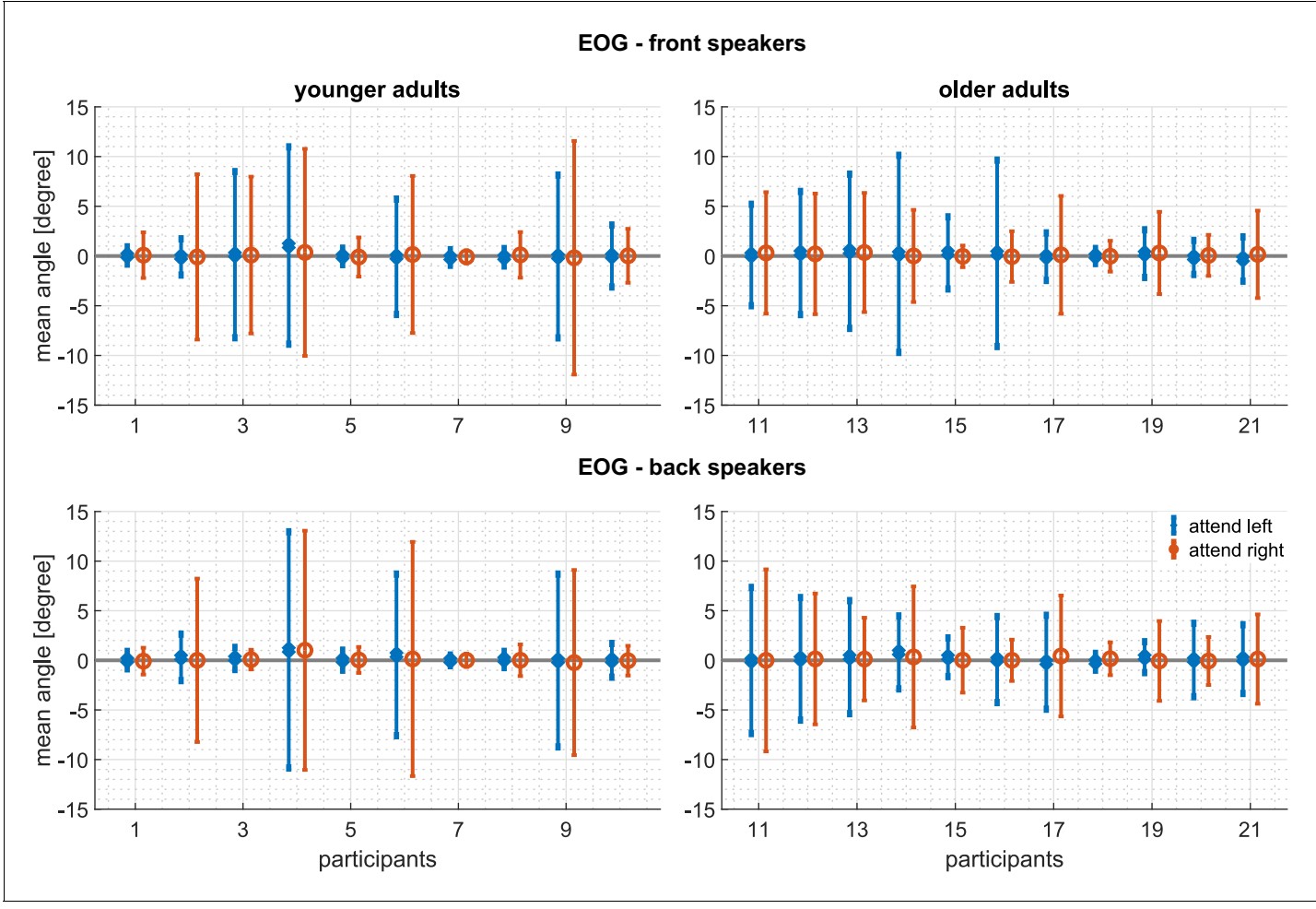

**Figure 8.** Experiment 2: Intrasubject variability of the horizontal EOG: Mean and standard deviations of the EOG during the complete trial of every participant. Top panels: attending the front speakers. Bottom panels: attending the back speakers. Left panels: younger adults. Right panels: older adults. Blue represents the EOG when attending the left, red when attending the right speaker. A positive EOG indicates that the gaze is directed toward the side of the attended speaker. Note that the deviation of the mean from 0° is well within one standard deviation and therefore indicates that participants did not systematically divert their gaze to the attended speaker. The following figure supplements are available for *Figure 8—figure supplement 1*. Time-resolved EOG analysis in Experiment 2; *Figure 8—figure supplement 2*. Activity of the frontalis and zygomaticus muscle in Experiment 2.

The online version of this article includes the following figure supplement(s) for figure 8:

**Figure supplement 1.** Time-resolved activity (sampling points represent the energy within consecutive 10 s segments) after pooling the ipsi- and contralateral EOG signals for the front (left) and back speakers (right) in Experiment 2.

**Figure supplement 2.** Grand average of the zygomaticus and frontalis muscle activity when stories were played from the front (top row) and back speakers (bottom row) in Experiment 2.

Portions of neocortex also play a role in controlling pinna movements. Lesions of auditory cortex reduce the kinematic complexity of pinna orienting and slow its habituation, see *Alvarado and Santibañez, 1971*. Stimulation and recording studies in the macaque have identified a premotor ear-eye field (area 8B), which is connected with both auditory cortical areas and the SC, see *Lanzilotto et al., 2013*. These animal neuroanatomy and physiology studies, coupled with *Wilson, 1908* seminal report, make it clear that the eyes and pinnae work together during endogenously cued attentional orienting.

## Future work

A recent study in humans by *Gruters et al., 2018*, showed a close relationship between movements of the left and right eardrums and multiple parameters of task-related left- and right-directed eye

movements. It will be important in future research to test whether muscular responses of the middle and outer ears are linked in a coordinated manner to ocular orienting. Furthermore, exploration of the relationship between human auriculomotor activity and subtle markers of covert attention (see the recent review given in *van Ede et al., 2019*) corticofugal modulation of ascending auditory pathways (*Perrot et al., 2006*) in endogenous attention, and the neural mechanisms of orienting discussed in the preceding section has scarcely begun.

Our results have implications for applied science, as well. They suggest that patterns of auricular muscle activity might serve as an easily accessible correlate of top-down processing in endogenous modes of attention. As such, the described effects might complement electroencephalographic indices of attentional focus (*de Cheveigné et al., 2018*; *Schäfer et al., 2018*) in that their sensitivity is exclusively spatial, rather than reflecting a context-specific mixture of modality, feature, location, and object representations. Registration of pinna-orienting might better support near real-time decoding of the attentional focus and, as compared to EEG-based stimulus reconstruction approaches, does not require the exogenous sound source, for example see the discussion in *Schäfer et al., 2018*. Thus, auricular muscle monitoring might support the decoding of auditory attention in technical applications such as attentionally controlled hearing aids that preferentially amplify sounds the user is attempting to listen to. We wish to underscore, though, that the development of such applications would benefit crucially from a better understanding of how the auditory and visual attention systems interact. We hope that the results presented here will stimulate research in this direction.

## Materials and methods

### Participants

Both older (N = 12, mean age = 62.7 ± 5.9 y, 8 F, all right-handed) and younger adult (N = 16, mean age = 24.1 ± 3.1 y, 8 F, 15 right-handed, one left-handed) volunteers in Experiment one had age-typical, pure tone audiometric thresholds (1, 2, 4, and 8 kHz; young < 20 dB; old < 40 dB). All served in both studies, but after the 8th participant, Experiment two was altered in several ways (e.g., four stimulus directions rather than two). Only data from the final 21 subjects were retained for Experiment 2. The two groups in this experiment comprised 11 older adults (mean age = 62.6 ± 6.2 y, 8 F, all right-handed) and 10 younger adult (mean age = 24.1 ± 3.6 y, 5 F, nine right-handed, one left-handed). After a detailed explanation of the procedure, all subjects signed a consent form. The study was approved by the responsible ethics committee (ethics commission at the Ärztekammer des Saarlandes, Saarbrücken, Germany; Identification Number: 79/16).

### Stimuli and tasks

The four active loudspeakers (KH120A, Neumann, Germany) were positioned at head level, 115 cm. Sounds in Experiment 1 and 2 were reproduced with a soundcard (Scarlett 18i20, Focusrite, UK). The experimental paradigms were programmed using software for scientific computing (Matlab, Mathworks, USA) and Psychtoolbox 3. In Experiment 1, sounds lasted 1.7 – 10.0 s, were delivered every 15 – 40 s, and had an average intensity of 70 dBC, except for foot steps (65 dBC). Each of the nine stimuli (lemur howling, dog barking, helicopter flying, cell phone vibrating, birds singing, baby crying, mosquito buzzing, footsteps, and traffic jam) was repeated four times (i.e., once per speaker). In Experiment 2, the stories were 5 min long, with an average intensity of 50 dBA for younger and 60 dBA for older participants. Participants answered content questions at the conclusion of each condition in this experiment.

### Electrophysiological recordings and signal processing

Surface EMGs were recorded with non-recessed, Ag/AgCl electrodes (BME4, BioMed Electrodes, USA), which were 4 mm in diameter for TAM and 6 mm (BME6) in all other cases, see *Figure 1*. The signals were AD-converted at 9600 Hz and 24 bit resolution per channel (4 × USBamp, g.tec GmbH, Austria). Skin temperature, skin resistance, electrocardiograms, and EOGs were also recorded. All signal processing algorithms were implemented using the scientific computing software Matlab (Mathworks, USA, Version: 2018a). Because surface electrodes had not previously been used to record from intrinsic ear muscles, we conducted preliminary tests with a participant who exhibited a

large, reliable Wilson's phenomenon and who could voluntarily contract her SAM and PAM. Isolation of the corresponding responses indicated that EMG from TAM electrodes was not an artifact of volume conduction from PAM or SAM. In other words, the TAM activity was not correlated with forced SAM/PAM innervation. Sternocleidomastoid EMG signals were zero-phase bandpass filtered from 60 to 1000 Hz (FIR, 2000th order), the auricular EMG signals from 10 to 1000 Hz (FIR, 2000th order) with a notch filter at 50 Hz (IIR, 2nd order). Horizontal EOG signals were zero-phase filtered from 0.01 to 20 Hz (IIR, 2nd order). All filter operations were performed using Matlab's filtfilt-function for zero-phase filtering. The filtered signals were then downsampled to 2400 Hz for further processing. The statistical analysis was performed using repeated measures ANOVA (with IBM SPSS Statistics 26). Within-subjects factors were stimulus-muscle correspondence (ipsi- vs. contralateral responses) and anteriority (front vs. back speakers). The only between-subjects factor our statistical model accounted for was age and, in association with that, also stimulus level in the endogenous experiment. Other factors like head-size, audiogram shape or small electrode placement differences were not included in the model. All main and interaction effects were tested.

## Exogeneous (transient) data

Root-mean-square (RMS) envelopes of the filtered and downsampled EMG signals were calculated with a sliding window (step size = 1 sample, window length = 150 samples/62.5 ms.) The data were then segmented into epochs extending from 3 s prior to stimulus onset until 3 s following termination of the auditory stimulus, which was of variable duration. Epochs were baseline corrected with respect to the mean RMS envelope amplitude of the pre-stimulus interval, that is this mean was subtracted from the epoch. For every participant, normalization was performed within every monitored auricular muscle (e.g., left PAM, right SAM) and for a specific stimulus type (e.g., traffic jam). Since every stimulus type was repeated four times, the largest voltage in the corresponding four epochs was used for normalization. The reference value in each case was the largest voltage at any time point within the four relevant epochs (e.g., the four directions/trials with a traffic jam stimulus) in the EMG recordings of that particular muscle. Each participant's data were pooled according to whether the side of the stimulus and recorded muscle did or did not match, and then were averaged into contralateral and ipsilateral waveforms. Eighteen trials contributed to each of these per subject contralateral and ipsilateral waveforms, nine from the left speaker and nine from the right. Mean amplitudes were computed across a measurement window extending from 100 to 1500 ms following stimulus onset and were then subjected to statistical analysis.

## Endogenous (sustained) data

Artifacts of the filtered and downsampled EMG data were reduced by averaging the signal energy of 1 s, non-overlapping segments and rejecting segments that deviated by more than two standard deviations from the mean. For each participant, the mean energy of a given channel during the four listening conditions (left/right × front/back) was calculated and then normalized to the largest value across the 5-min run. These normalized data were then averaged into ipsilateral/contralateral categories and subjected to statistical analysis.

## EMG time-frequency decomposition (for *Figure 5—figure supplements 1* and *2*)

As the rather sustained muscle activity during endogenous attention might be reflected in low frequency components according to convolution models of the EMG (*Farina et al., 2014*), filtered and downsampled EMG signals in Experiment two were decomposed into eight frequency bands by a nonsubsampled octave-band filter bank (5th order Daubechies filter). Each frequency band was then further processed in the same fashion as the broadband signals reported in the main text.

## Computer vision setup and motion analysis (for *Videos 1*, *2* and *3*)

Videos were acquired using four Ximea MQ022CG–CM color sensors with a resolution of 1936 × 1216 at 120 frames per second and an exposure of 2 ms. Two cameras were positioned on each side of the head and focused on the ears to record pairwise stereo videos. We used hardware triggering for all four cameras and recorded each camera onto a separate m.2 solid-state-drive to reduce frame loss. We used a KOWA 35 mm macro lens with an aperture of F0.4 which gave us a

close-up view of the ear with acceptable depth of field to allow slight movements towards the camera and enough distance such that the cameras did not cast shadows on to the scene. We illuminated the face uniformly with flicker-free LED studio illumination. The cameras were calibrated with the stereo camera calibrator app from the Mathworks Matlab Computer Vision System Toolbox. Calibration was performed whenever camera adjustment required re-alignment of relative stereo camera positions or a change of focus of one of the cameras.

For 3D reconstruction and motion visualization/quantification, we used functions from the Mathworks Matlab Computer Vision System Toolbox and custom written code. Our analysis system was able to reduce redundancies in optic flow and stereo depth estimation by exploiting the unilateral scene composition and limited degrees of freedom for ear and head movements. For 3D reconstructions, we initialized a sequence with one initial estimation of disparity and subsequently tracked points independently for the left and right image sequence.

We tracked points with respect to the first frame of the sequence as reference frame with dense optical flow initialized with a rigid motion estimation. Motion was visualized with a Lagrangian motion magnification approach that had a constant magnification factor with respect to the reference frame and prior removal of affine motion with respect to manually selected stable points. The results of the motion analysis with and without magnification can be seen in the videos.

## Acknowledgements

This study was partially supported by the German Federal Ministry of Education and Research, Grant No. BMBF-FZ 03FH004IX5 (PI: DJS). We thank Larissa Arand for assistance with data collection and Becca Sullinger for the artwork in *Figure 1*. We acknowledge support by the Deutsche Forschungsgemeinschaft (DFG, German Research Foundation) and Saarland University within the funding programme Open Access Publishing.

## Additional information

### Funding

| Funder | Grant reference number | Author |
| --- | --- | --- |
| Bundesministerium für Bildung und Forschung | BMBF-FZ No. 03FH004IX5 | Daniel J Strauss |

The funders had no role in study design, data collection and interpretation, or the decision to submit the work for publication.

### Author contributions

Daniel J Strauss, Conceptualization, Resources, Formal analysis, Supervision, Validation, Investigation, Methodology, Writing - original draft, Project administration, Writing - review and editing; Farah I Corona-Strauss, Conceptualization, Software, Formal analysis, Supervision, Validation, Investigation, Visualization, Methodology; Andreas Schroeer, Software, Formal analysis, Validation, Investigation, Visualization; Philipp Flotho, Data curation, Software, Formal analysis, Validation, Investigation, Visualization; Ronny Hannemann, Conceptualization, Methodology; Steven A Hackley, Conceptualization, Formal analysis, Investigation, Methodology, Writing - original draft, Writing - review and editing

### Author ORCIDs

Daniel J Strauss https://orcid.org/0000-0001-8481-499X
Andreas Schroeer http://orcid.org/0000-0002-7904-3622

### Ethics

Human subjects: The study was approved by the responsible ethics committee (ethics commission at the Ärztekammer des Saarlandes, Saarbrücken, Germany; app. number 79/16) After a detailed explanation of the procedure, all subjects signed a consent form.

**Decision letter and Author response**
Decision letter https://doi.org/10.7554/eLife.54536.sa1
Author response https://doi.org/10.7554/eLife.54536.sa2

## Additional files

### Supplementary files
• Supplementary file 1. Table Supplements for Experiment 2.
• Transparent reporting form

### Data availability

The code with and data are available from GitHub https://github.com/a-schroeer/ExogEndogEarProject (copy archived at https://github.com/elifesciences-publications/ExogEndogEarProject) and the Dryad Digital Repository https://doi.org/10.5061/dryad.d4md86r, respectively.

The following dataset was generated:

| Author(s) | Year | Dataset title | Dataset URL | Database and Identifier |
|---|---|---|---|---|
| Strauss DJ, Corona-Strauss FI, Schroeer A, Flotho P, Hannemann R, Hackley SA | 2020 | Data from: Vestigial auriculomotor activity indicates the direction of auditory attention in humans | https://doi.org/10.5061/dryad.d4md86r | Dryad Digital Repository, 10.5061/dryad.d4md86r |

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
