## [Decision Letter]

**Acceptance summary:**

Unlike cats and dogs, humans can at most barely move their ears. This study reports that the largely vestigial muscles of the ear nevertheless retain a sensitivity to spatial attention: changes in electrical activity of ear muscles could be evoked both in subjects deliberately attempting to listen to a story from one location while ignoring another, and when subjects were surprised by novel sounds while reading an irrelevant essay. The attentional effects on ear muscle electrical activity reported in this study substantially expand the scope of knowledge regarding top down mechanisms in the brain and how they influence the sense organs.

**Decision letter after peer review:**

[Editors’ note: the authors submitted for reconsideration following the decision after peer review. What follows is the decision letter after the first round of review.]

Thank you for submitting your work entitled "Vestigial auriculomotor activity indicates the direction of auditory attention in humans" for consideration by *eLife*. Your article has been reviewed by three peer reviewers, and the evaluation has been overseen by a Reviewing Editor and a Senior Editor. The following individuals involved in review of your submission have agreed to reveal their identity: Christopher Shera (Reviewer #1); Sarah Verhurst (Reviewer #2).

Our decision has been reached after consultation between the reviewers. Based on these discussions and the individual reviews below, we regret to inform you that this submission of your work will not be considered further for publication in *eLife*. However, the reviewers found merit in the project. Should you choose to revise the manuscript, we ask that authors include the original submission manuscript number, title, and corresponding author and upload a point-by-point response to the reviews including any changes they have made to the manuscript.

The comments of the reviewers are included below. As you may be aware, at *eLife* those initial reviews set the stage for a subsequent consultation among the reviewers to achieve consensus. Most of our discussion centered on the most challenging comments from reviewer 3 regarding the relationship between your findings and eye movements. A possible relationship to eye movements would be interesting either way, but much of this material is only provided in the supplementary figures, and lacks narrative description to make the implications clear (in particular the series of orange and black figures in the supplementary materials).

We recognize that EOG recordings provide a limited resolution that likely preclude looking at microsaccades in Experiment 2, but Experiment 1 provides an opportunity to ascertain whether there is any ear muscle activity associated with the macrosaccades that occur during reading.

We also suggest that the results concerning the frontal speakers that are currently presented only in the supplementary materials be included in the main manuscript. As there is no obvious conceptual reason to discount the results from frontal space, they should be given equal prominence as the results from rear space.

Reviewer #1:

The manuscript is generally well written, the experiments thorough and compelling, and the content novel and genuinely intriguing. My comments are relatively minor.

Abstract: The logic of the first paragraph of the Abstract is confusing and the phrasing sometimes awkward. I suggest reworking it along these lines:

"Humans, unlike dogs and cats, are not commonly thought to move their ears when focusing auditory attention, either reflexively toward novel sounds or voluntarily toward those that are goal-relevant. Nevertheless, humans may retain a vestigial pinna-orienting system that persists as a "neural fossil" within the brain. Consistent with this hypothesis, we demonstrate that the direction of auditory attention is reflected in the sustained electrical activity of muscles within the vestigial auriculomotor system."

Results paragraph two: What is ps? A typo? Or is it somehow supposed to be the plural of p? If the latter, it would be much clearer to write "p values < 0.001".

Results paragraph four: This paragraph is out of place in the Results and should be moved to the Discussion.

Final paragraph of the Results: This paragraph is also rather jarringly out of place and should be moved to the Discussion.

In the same paragraph: Citation needed for "found in multiple languages".

Discussion paragraph two: This might be a good place to include the phrase "about 25 million years ago" which was lost in the rewrite of the Abstract.

In the same paragraph: Should be "New World monkeys" and "Old World monkeys"

Subsection “Exogeneous (transient) data”: What was the overlap between consecutive windows (step size)?

Discussion: The manuscript should cite and discuss the recent and likely related work of Gruters et al., 2018.

Reviewer #2:

The manuscript presents how auriculomotor activity forms an objective correlate of exogenous and endogenous auditory attention. To this end, study participants had their heads in a fixed position while listening to sounds coming from different directions to steer auditory attention. At the same time, TAM, PAM, SAM and AAM EMG muscle activity and pinna movements were measured, the latter with a camera. The study is convincing in relating the exogenous auriculomotor activity (increase of PAM/AAM) to auditory attention, because SCM and EOG signals were used to rule out other explanatory variables such as eye gaze and horizontal movement of the head. The presented analysis and supplementary material support the main conclusions. I have one major point related to data-processing, and others are mostly related to interpretation/applicability of results. I am expecting that the data-processing point will not change the main outcome of the study.

1) I have difficulties understanding the normalization procedure described in the text and its relationship to the values labeled on Figure 2.

"Epochs were baseline corrected..[].. Amplitudes were normalized separately for each participant and muscle according to the largest value among the four stimulus presentations at any time point within the epoch data"

When reading this, I interpret that for each person, the peak amplitude of the signal in the largest condition should be one (i.e. normalized), and that in the other three conditions for this person the amplitude should be less than one. Then data was averaged across conditions, after which a mean amplitude across a window of 1400 ms was calculated, or pooled across participants to yield the grand average waveform. When looking at Figure 2, the amplitudes have amplitudes of 100 to 125 [-], and I do not follow the relationship between those numbers and the description in the text. Also, in case magnitudes are normalized to peak maxima, I would expect quite a variability in the baselines of different individuals which should show up strongly in the grand-averaging across people with different base-line estimates.

2) The characterization of the vestigial network was performed on the basis of a still head during the task, which was necessary to demonstrate the main point of the paper. However, this study does not really go into whether and how strongly this auriculomotor activity plays when people are allowed to move their heads during an attention task. i.e., would this mechanism be complementary to attention-driven gaze, movement steering or does only occur when the head itself cannot move? This differentiation might be important to consider when translating this work to hearing-aid applications. This point is not a drawback of the paper, but should perhaps be discussed more strongly when discussing the potential application areas.

Reviewer #3:

General assessment: There are some aspects of the paper that I found intriguing, but there are a number of points that I found either under-explored, or unconvincing. A stronger mechanistic case should also be made relating these findings to others in the literature. The case for using this to aid decoding is also weak. Ultimately, I find that this article falls (fairly far) below the standard of what I would expect for *eLife*.

Substantive concerns.

1) The authors could do a much better job placing the current results in the context of other subtle indicators of covert attention. There is an extensive literature on any number of subtle indicators of covert attention (e.g., microsaccades, pupil dilation, even subtle levels of neck muscle recruitment; see van Ede Chekroud Nobre, Nat Human Behavior, 2019 for a recent article in this field), and mechanistic evidence tying these to the superior colliculus. Given that the authors invoke the superior colliculus as a possible node within the auriculomotor pathway, it would help to speculate on how the current results fit in with other work in this literature.

2) Consideration of these other measures leads to concerns about discounting the possibility of subtle eye or head movements. For eye movements, the use of electrooculography to address gaze orientation is not sufficient. EOG has good temporal resolution, but its spatial resolution is very poor, and can't be used to rule out anything with saccadic amplitudes less than 1-2 deg. Further, given recent results linking eye movements to movements of the eardrum in humans and monkeys (Gruters et al., 2018), a much more precise linking of eye movements are auricular muscle recruitment is warranted, and this could be done much more systematically (e.g., are the auricular muscles actually recruited during eye movements?). Discounting head movements using surface EMG recordings of SCM is also insufficient for a number of reasons. As a powerful head turning muscle, SCM tends not to be recruited for subtle movements of the head. Further, SCM contributes to contralateral, not ipsilateral, head turns, so the focus on the ipsilateral SCM muscle in Experiment 1 supplementary figure 5 is incorrect. Overall, I found the measures used to discount the possibility of subtle eye or head movements to be unconvincing.

3) The authors speculate that signals from the auricular muscles could be used to decode the locus of auditory spatial attention in near real time. While of potential interest, this claim is highly speculative given the coarseness and apparent variability in the signals shown in the manuscript (which generally show grand averages, with little to no sense of variability). If the authors wish to make the decoding argument, then why not try this? How well can target location actually be extracted from the current data? This would seem to be a tractable question for Experiment 1 (e.g., use data from some subset of trials to train a classifier, and then see how well the classifier works on the other set of trials). Chance performance would be 25% -- can a classifier based on auricular muscle activity do substantially better? I must admit that I am sceptical that signals extracted from these small signals could be useful at all in the real world, given how much the ears move during facial expressions or voluntary ear wiggling. Unless the decoding case can be made more strongly, my advice would be to drop the "decoding" angle from the paper and focus on basic findings.

4) For Experiment 2, EMG activity is basically averaged across the entire 5 min range. This is a very coarse timeframe and approach, and I can't help but think there would be something more interesting in the data. Is there any way of looking for transient changes in auricular muscle recruitment, and then tying that back to some sort of event during the stream of auditory information? There is the chance of some potentially rich data that really hasn't been mined with the current approach.

[Editors’ note: further revisions were suggested prior to acceptance, as described below.]

Thank you for resubmitting your work entitled "Vestigial Auriculomotor Activity Indicates the Direction of Auditory Attention in Humans" for further consideration by *eLife*. Your revised article has been evaluated by Barbara Shinn-Cunningham (Senior Editor) and a Reviewing Editor.

The manuscript has been improved and all three reviewers appreciate the importance of this work, but there are some remaining issues that need to be addressed. In particular, reviewer 3 's concerns center on the eye movement results as well as several issues concerning head and neck musculature. These concerns will need to be addressed before the paper can be accepted. All three reviewers and reviewing editor have consulted and agree on the importance of incorporating these additional analyses. The full reviews are included below.

Reviewer #1:

I carefully read both rebuttal letter and the revised manuscript and in my view, the revised manuscript is very clear, original and of high scientific standard. The added section on auditory-visual interactions and expanded discussion have turned this paper into a very nice and complete paper.

Reviewer #2:

The authors have generally done a fine job addressing the reviewer concerns. …

Reviewer #3:

General assessment

This short manuscript reports that tasks that engage auditory attention either exogenously (Experiment 1) or endogenously (Experiment 2) lead to the recruitment of auricular muscles that subtly change the shape of the pinna, doing so in a spatially-dependent manner. The conclusion is that such recruitment attests to the presence of a vestigial brain circuit. The topic is timely given recent findings linking saccadic eye movements to movements of the eardrum, and a number of other subtle indicators of covert attention driven, for example, by the oculomotor system. The results are intriguing, but more can be done to address other potential confounds, particularly on the oculomotor side.

Substantive concerns

The authors have extensively revised the manuscript, and established the phenomena both within and across their subject pool. I still have some concerns about other potential confounds that need to be addressed; as the authors say the neural drive to the ear muscles is so weak that the resultant movements are miniscule compared to those generated during broad smiles or wiggling.

1) Previously, I had raised concerns about potential confounds from the oculomotor system that orients the line of sight via eye and/or head movements (what the authors term the "visuomotor system" in their response). The authors have added a number of analyses that go some to length to assuage concerns about eye movements. However, grand average measures of EOG across many trials could mask some interactions between eye movements and auricular muscle activity; the data shown in Figure 7 also shows how the variance of the EOG signal decreases after stimulus onset, particularly for stimuli presented at the left-back speaker. More analyses and details are warranted.

1a) For Experiment 1, the auditory stimuli are presented while subjects are "reading a boring essay". Please provide details about how large the eye movement excursions were; from Multimodal Figure supplement 1, it appears that the text spanned about +/-12 deg of horizontal visual angle, but this is just one subject.

1b) The authors acknowledge that they can detect "macro" saccades greater than about 1 degree on average, and these should be analyzed in a more systematic manner than relying on average EOG traces, which could wash out effects. The oculomotor literature on microsaccades has a number of ways of presenting spatial and temporal patterns of saccades timed to external events (e.g., see saccadic “rasters” in Figure 4 of Tian, Yoshida and Hafed Front Syst Neurosci 2016), and I think these should be applied to the data from Experiment 1. See for example the work by Ziad Hafed (Figure 4 in Tian, Yoshida and Hafed Front Syst Neurosci 2016). Something similar for the "macrosaccade" data from Experiment 1 could is needed to establish whether or not stimulus onset is altering the patterning of larger saccadic eye movements.

1c) For Experiment 2, please provide the "time-resolved" plots for EOG data, similar to what is provided for auricular muscle activity in Figure 6. The EOG data shown in Figure 8 is helpful, but shows data averaged across an entire 5 min segment I believe, which is a very large window.

2) In regards to a potential concern about head movements, I agree that the vestibular-auricular reflex is unlikely, given that the head was stabilized on a chin-rest. My previous concern was more about whether the act of spatially deploying auditory attention was related to neck muscle contraction that introduced cross-talk at the auricular muscles (the absence of head movement can't be used to infer the absence of neck muscle contraction in this regard). The work by Cooke and Patuzzi, 2014, doesn't address this concern since they examined sternocleidomastoid activity during ipsilateral head turns (right PAM and right SCM recordings during right head turns). SCM is a contralateral, not ipsilateral head turner, so would be directly recruited by leftward turns, which do not appear to have been studied in the Cooke and Patuzzi setup. The Cooke and Patuzzi paper actually mentions the concerns I have about potential cross-talk from other nearby muscles (see end of the first paragraph of their "subjects and methods"). Muscles on the back of the neck (e.g., splenius capitis or suboccipital muscles; insertion on the occiput, which lies close to the mastoid) are ipsilateral head turners. The activity of the suboccipital muscles in particular has also been related to reflexive visuospatial attention in a Posner type task in head-restrained monkeys (e.g., see Corneil et al., 2008). To be clear, I don't think that the entirety of the results could be "explained" by cross-talk from nearby dorsal neck muscles, but the authors should consider this perspective.

3) A final point about other muscles; Figure 1 shows that the authors recorded EMG activity on zygomaticus major and frontalis, but results are not analyzed. Given the point about how small the movements of interest are compared to those related to smiling, please establish the independence of the auricular muscle recordings from these facial muscles.

---

## [Author Response]

[Editors’ note: the authors resubmitted a revised version of the paper for consideration. What follows is the authors’ response to the first round of review.]

Reviewer #1:The manuscript is generally well written, the experiments thorough and compelling, and the content novel and genuinely intriguing. My comments are relatively minor.Abstract: The logic of the first paragraph of the Abstract is confusing and the phrasing sometimes awkward. I suggest reworking it along these lines:"Humans, unlike dogs and cats, are not commonly thought to move their ears when focusing auditory attention, either reflexively toward novel sounds or voluntarily toward those that are goal-relevant. Nevertheless, humans may retain a vestigial pinna-orienting system that persists as a "neural fossil" within the brain. Consistent with this hypothesis, we demonstrate that the direction of auditory attention is reflected in the sustained electrical activity of muscles within the vestigial auriculomotor system."

Thank you very much for disentangling and sharpening this paragraph. We moved your suggestions 1:1 to the revised version of the manuscript (we just added “for about 25 million years”).

Results paragraph two: What is ps? A typo? Or is it somehow supposed to be the plural of p? If the latter, it would be much clearer to write "p values < 0.001".

An anecdote: The “ps” was causing a discussion among the authors before the first submission. Some liked it as it is very common in psychology, some disliked it as it is horrible from mathematical point of view. We followed your advice and used “p-values” in the revised version of the manuscript.

Results paragraph four: This paragraph is out of place in the Results and should be moved to the Discussion.

Thanks for the observation. We moved this paragraph along with some associated text to the Discussion.

Final paragraph of the Results: This paragraph is also rather jarringly out of place and should be moved to the Discussion.

Thanks for the observation. We deleted most of this paragraph and moved the rest to the Discussion.

In the same paragraph: Citation needed for "found in multiple languages".

In the interest of space, we have omitted the comments about ear-straining metaphors from the revised manuscript.

Discussion paragraph two: This might be a good place to include the phrase "about 25 million years ago" which was lost in the rewrite of the Abstract.

Included in the revised version. Thanks

In the same paragraph: Should be "New World monkeys" and "Old World monkeys"

Amended in the revised version. Thanks.

Subsection “Exogeneous (transient) data”: What was the overlap between consecutive windows (step size)?

The RMS value (windows size = 150 samples) was calculated for every sample without overlap or omission (so a step size of 1; that is why we refer to it as an “RMS-envelope”). This is more clearly stated in the revised version (Materials and methods subsection “Electrophysiological Recordings and Data Processing”).

Discussion: The manuscript should cite and discuss the recent and likely related work of Gruters et al., 2018.

We modified the Discussion accordingly, see general comments and comments to reviewer 3. This important citation is now included along with a more complete discussion of eareye interactions.

Reviewer #2:The manuscript presents how auriculomotor activity forms an objective correlate of exogenous and endogenous auditory attention. To this end, study participants had their heads in a fixed position while listening to sounds coming from different directions to steer auditory attention. At the same time, TAM, PAM, SAM and AAM EMG muscle activity and pinna movements were measured, the latter with a camera. The study is convincing in relating the exogenous auriculomotor activity (increase of PAM/AAM) to auditory attention, because SCM and EOG signals were used to rule out other explanatory variables such as eye gaze and horizontal movement of the head. The presented analysis and supplementary material support the main conclusions. I have one major point related to data-processing, and others are mostly related to interpretation/applicability of results. I am expecting that the data-processing point will not change the main outcome of the study.1) I have difficulties understanding the normalization procedure described in the text and its relationship to the values labeled on Figure 2."Epochs were baseline corrected..[].. Amplitudes were normalized separately for each participant and muscle according to the largest value among the four stimulus presentations at any time point within the epoch data"When reading this, I interpret that for each person, the peak amplitude of the signal in the largest condition should be one (i.e. normalized), and that in the other three conditions for this person the amplitude should be less than one. Then data was averaged across conditions, after which a mean amplitude across a window of 1400 ms was calculated, or pooled across participants to yield the grand average waveform. When looking at Figure 2, the amplitudes have amplitudes of 100 to 125 [-], and I do not follow the relationship between those numbers and the description in the text. Also, in case magnitudes are normalized to peak maxima, I would expect quite a variability in the baselines of different individuals which should show up strongly in the grand-averaging across people with different base-line estimates.

In the first step, every epoch was baseline corrected by subtracting the mean baseline value from the epoch. Then, normalization was performed, which you understood correctly: The largest (absolute) value in four conditions (for every stimulus type/subject) is set to 1 (or -1, if the baseline subtraction introduced negative values) and all other values are scaled accordingly. This normalization procedure should be clearer in the revised version. Along these lines, we also point out more clearly that the normalization is not only done independently for every subject and channel, but also for every stimulus type. For example, for subject x, all 4 responses to the stimulus “baby crying” (which was presented 4 times, once from each speaker) recorded at the same channel (for example right PAM) were normalized with respect to each other. The next stimulus, for instance, “dog barking”, was then processed independently. In the next phase, responses of the same stimulus type were pooled (and averaged) to form ipsi- or contralateral responses, defined according to lateral congruence of sound source and recorded muscle. (It was at this point that the orange-and-black epoch matrices presented in the supplement were generated). Then, responses were averaged for every subject, and mean values between 100-1500 ms were calculated for use in the statistical analyses.

Regarding the plot: For plotting purposes in the previous version of the manuscript, we increased the values to % baseline, which is why the baseline had an average of 100 in the plot. The initial idea was to make those plots more comparable to those in our Stekelenburg and van Boxtel, 2002, reference. However, we see that there was room for confusion, especially as this was not in line with the matrix plots in the supplementary material. Therefore, we removed the percentage-based scale for the y-axis in the revised version. Thanks for drawing our attention to this issue.

Regarding baseline variability: While there are cases in which peak normalization produces large baseline variability, averaging across epochs ameliorates this problem because the activity is not locked to any event. In the revised version, we have added single-epoch plots along with the average and standard deviation (see Figure 3 and related material in the supplement). Thus, it should be much easier to appreciate the variance within our data in the revised manuscript.

2) The characterization of the vestigial network was performed on the basis of a still head during the task, which was necessary to demonstrate the main point of the paper. However, this study does not really go into whether and how strongly this auriculomotor activity plays when people are allowed to move their heads during an attention task. i.e., would this mechanism be complementary to attention-driven gaze, movement steering or does only occur when the head itself cannot move? This differentiation might be important to consider when translating this work to hearing-aid applications. This point is not a drawback of the paper, but should perhaps be discussed more strongly when discussing the potential application areas.

This is an important insight, and one that we will be addressing in an upcoming paper focused on decoding attention direction from pinna EMG signals. Thank you very much for bringing this up, Dr. Verhulst. However, based on comments from another reviewer, we have decided to reduce discussion of hearing aids and other potential applications in the present manuscript. As you mentioned, it was our main goal in this initial study to isolate the auriculomotor effect as much as possible, so head-ear interactions were not emphasized. Neurobiological research in cats has begun to identify the ways in which pinna orienting is coordinated with head and eye movements (Populin and Yin, 1998). It seems to be more complicated than eye-head coordination due to the acoustic shadow of the head and multidimensionality of ear movements. We briefly allude to the topic in our discussion of the vestibulo-auricular response and future work.

Reviewer #3:General assessment: There are some aspects of the paper that I found intriguing, but there are a number of points that I found either under-explored, or unconvincing. A stronger mechanistic case should also be made relating these findings to others in the literature. The case for using this to aid decoding is also weak. Ultimately, I find that this article falls (fairly far) below the standard of what I would expect for eLife.

The authors would like to thank reviewer 3 for the critical feedback. Our initial submission was motivated by our finding that there is neural drive to auricular muscles when paying attention. We presented evidence in form of a short report which documented for the first time a direct spatial correspondence between sustained auditory attention and sustained auriculomotor activity, as measured electromyographically. The auriculomotor system as such is almost untouched in the literature. Indeed, all previous relevant work for “auriculomotor” was reviewed within the strict space limits of our short report submission. However, Reviewer 3 looked at the data from a new but important angle, focusing mainly on the interactions between the auditory and the visual motor systems. Even though we are convinced that our results are of merit focusing on the auditory modality (the 1^st^ and 2^nd^ reviewer are also positive in this sense), we completely agree that this new aspect is a very important component for the discussion of our results. This is especially true in light of the recent work linking eye to eardrum movements. Furthermore, we believe that a careful consideration of this topic, embedding our findings within the literature, will extend the study’s impact and more effectively stimulate further research. We clearly stated this now in “Future Work”.

Substantive concerns.1) The authors could do a much better job placing the current results in the context of other subtle indicators of covert attention. There is an extensive literature on any number of subtle indicators of covert attention (e.g., microsaccades, pupil dilation, even subtle levels of neck muscle recruitment; see van Ede Chekroud Nobre, Nat Human Behavior, 2019 for a recent article in this field), and mechanistic evidence tying these to the superior colliculus. Given that the authors invoke the superior colliculus as a possible node within the auriculomotor pathway, it would help to speculate on how the current results fit in with other work in this literature.

As we mentioned before, we completely agree with the reviewer regarding this critique. We have added a new paragraph in the Discussion section focusing on this topic.

2) Consideration of these other measures leads to concerns about discounting the possibility of subtle eye or head movements. For eye movements, the use of electrooculography to address gaze orientation is not sufficient. EOG has good temporal resolution, but its spatial resolution is very poor, and can't be used to rule out anything with saccadic amplitudes less than 1-2 deg. Further, given recent results linking eye movements to movements of the eardrum in humans and monkeys (Gruters et al., 2018), a much more precise linking of eye movements are auricular muscle recruitment is warranted, and this could be done much more systematically (e.g., are the auricular muscles actually recruited during eye movements?). Discounting head movements using surface EMG recordings of SCM is also insufficient for a number of reasons. As a powerful head turning muscle, SCM tends not to be recruited for subtle movements of the head. Further, SCM contributes to contralateral, not ipsilateral, head turns, so the focus on the ipsilateral SCM muscle in Experiment 1 supplementary figure 5 is incorrect. Overall, I found the measures used to discount the possibility of subtle eye or head movements to be unconvincing.

The focus of the initial submission as short report was the analysis of auricular muscle activation during spatial auditory attention and not the analysis of the interaction between the auditory and visual systems per se. We merely wanted to monitor relationships between visual and auditory motor systems that are currently known to exist: the oculo-auricular phenomenon of Wilson. In this way, we could exclude that the described effect is secondary to the Wilson’s phenomenon (as we understand it today) or to gross neck movements. Note that the latter has also been ruled out earlier (see Cooke and Patuzzi, 2014). We did not plan our study or look at the data/results from the new but important angle recommended by the third reviewer—that of stressing possible aspects of interactions between the auditory and the visual motor systems that might beyond Wilson’s phenomenon. Accepting the reviewer’s critique and following also the editors’ advice, we analyzed and discussed much more carefully the interactions between the visual and the auditory systems reflected in our data. We also stated clearly that our setup only allows us to rule out spatial attention effects that are secondary to Wilson’s phenomenon. In particular, it becomes clear that:

1) Grand-average waveforms (shown in the supplementary information, Experiment 1) show a complete absence of location-specific eye movements prior to or synchronous with auricular responses. This does not rule out an effect of eye movements that were too small to be recorded with EOG which, as the reviewer notes, has a resolution of about 2 degrees of arc.

2) However, Wilson’s phenomenon is rarely elicited by eye movements less than 30 degrees (Urban, Marczynski, and Hopf, 1993). Furthermore, the enhancement of PAM activity appears to become laterally specific (ipsi > contra) only for gaze shifts greater than about 40 degrees (Patuzzi and O’Beirne, 1999, Figure 4). Our EOG recordings were certainly adequate for detecting ocular movements of this size.

3) There were large saccades in Experiment 2 as participants moved from the right edge of one line of text to the left edge of the next line of text. Inspection of single epochs failed to identify any systematic association between PAM responses and large saccades, see Author response image 1 for four representative trials.

**Author response image 1. sa2fig1:** Macrosaccades in the EOG (black line) during reading in Experiment 1 for subject 15 as example (a subject with large, clear PAM activations). It is noticeable that the muscle activations are not linked to the macrosaccades.

4) Activation of TAM during Wilson’s phenomenon exhibits lateral asymmetry in the direction opposite to that of our findings. Visible movements (Gertle and Wilkinson, 1929) and EMG activity (Urban et al., 1993) for this muscle are greater on the side opposite to the direction of gaze. By contrast, in our study TAM activation was enhanced when attention was elicited by sounds on the same side as the muscle.

5) Only the very fastest saccades (Fischer and Ramsperger, 1984) have a latency comparable to that of the auricular responses documented by our study (70 ms).

6) The latency of TAM responses with respect to onset of the eye movement in Wilson’s phenomenon averages about 340 ms (Schmidt and Thoden, 1978), far too slow to contribute to the attention effects we have documented in Experiment 1.

Nonetheless, we clearly state now in the revision that our experiment is not designed to discover or analyze interactions beyond of what is currently known about Wilson’s phenomenon in ear-eye interaction (i.e., that it is slow and only occurs during large, sustained gaze shifts). Especially in the light of recent of work of Jennifer Groh’s group, there is a lot of room for future investigations on this topic. We hope that our results help to stimulate such research.

3) The authors speculate that signals from the auricular muscles could be used to decode the locus of auditory spatial attention in near real time. While of potential interest, this claim is highly speculative given the coarseness and apparent variability in the signals shown in the manuscript (which generally show grand averages, with little to no sense of variability). If the authors wish to make the decoding argument, then why not try this? How well can target location actually be extracted from the current data? This would seem to be a tractable question for Experiment 1 (e.g., use data from some subset of trials to train a classifier, and then see how well the classifier works on the other set of trials). Chance performance would be 25% -- can a classifier based on auricular muscle activity do substantially better? I must admit that I am sceptical that signals extracted from these small signals could be useful at all in the real world, given how much the ears move during facial expressions or voluntary ear wiggling. Unless the decoding case can be made more strongly, my advice would be to drop the "decoding" angle from the paper and focus on basic findings.

Because of the current interest in decoding spatial auditory attention from the EEG signal, the authors thought that this might be an interesting application of the proposed auriculomotor monitoring. Apart from the stimulus reconstruction approach in which EEG signals are employed to reconstruct the envelope of the attended speaker (identification of attention direction requires correlating EEG and speech envelopes), the decoding of endogenous spatial attention from the EEG only (i.e., without the speech envelope) still remains a challenge. One could assume that an activation of the auricular muscles due to spatial attention might “amplify” the endogenous signal. In fact, originally, we found the demonstrated effect of the auricular muscle activation in endogenous modes of attention by trying to decode spatial auditory attention from the EEG (in the BMBF Attentional Microphone project; PI: DJS). A robust effect was just identified for the mastoid electrodes for larger frequencies which turned out to be related to the PAM activation. Driven by this, we just took the EMG with carefully attached electrodes and combined a hybrid machine learning scheme developed before in DJSs group (Strauss and Steidl, JCAM, 2002) to decode spatial auditory attention. For left/right decisions, the decoding scheme reached a performance a way above chance level (abstract at IEEE EMBC 2018 and SPR 2018). However, instead of using a black box learning scheme with abstract EMG features, we were interested in quantifying and analyzing the regularities between spatial auditory attention and the auricular EMG and, perhaps, associated pinna movements. This was the motivation for the present study. The experiments are not designed for a machine learning-based decoding in which many other factors matter, such as recording time and the associated electrodeskin interface stability. We agree with the reviewer that we should remove the engineering application / decoding from the paper as this is indeed not the subject here. In the revised version, we only mention a possible decoding application briefly in the discussion to stimulate a possible interested in those who work on EEG based decoding.

However, just to complete this response, we would like show that decoding the left/right listening direction from the back speakers (similar as in the EEG literature; a related setup was used in Schäfer et al., 2018) is possible with the described data. In particular, we demonstrate the performance of 2 different endogenous attention decoding approaches in Author response image 2 and Author response image 3 (i.e., for Experiment 2).

Here Author response image 2 shows the application of a hybrid wavelet-support vector machine classification of waveforms (Strauss and Steidl, “Hybrid Wavelet-Support Vector Machine Classification of Waveforms”, J of Comput and Appl. Math 2002) of the right/left listening direction jointly for the PAM and SAM. This machine is individualized (learned) for each participant and side of the head/ear using EMG data segments of 1s. If the (trained) machines on both sides give the same output at segment n, the corresponding direction is decoded. Otherwise the decoding system is in a doubt state and there is no decision in the decoded direction (i.e., the decoding scheme outputs the same the direction that it had at n-1, i.e., its previous state). For this simple test the filter bank was not adapted, nor the hyperparameters of the support vector machine were particularly tuned. The training set consisted of 162 observations (based on K-nearest-neighbor to mean) and the independent test set of 226 observations with a standard deviation of 32 (because of an energy-threshold based artefact rejection) for each of the 21 participants in Experiment 2. It is easy to see that the performance with >90 % is far above chance level (50%). The technical details of such an individualized decoding scheme with hybrid kernel learning machines for hearing aids can be found in Corona-Strauss, Hannemann, Strauss. Method for Operating a Hearing Aid Device. US Patent Application # 16102983 (Priority date: 14.08.2017 from the German application # 102017214163.8).

Apart from these adaptive concepts, we also evaluated the analysis of the mean energy (broadband signal) described in the paper as feature for a support vector classifier by means of a “leave-oneparticipant-out” cross validation: That is, the machine is trained with N-1 participants (N=21 in Experiment 2) and participant #N (which was not part of the training set) is classified; then shuffled such that a new set of N-1 participants is used for training and another participant #N-1 is classified (i.e., one participant serves as test set). As we have already mentioned, the mean energy is used as feature as in the time resolved analysis in Figure 6 but with segments of 1s (instead of 10s in the paper) to provide near real time decoding. Note that here we applied a learning across participants without individualization (as in in the analysis in Author response image 2). 10602+/-21 data segments/observations were used for training (from 20 participants) and an independent test set of 530+/-21 from 1 participant. Author response image 3 shows the result for the 21 independently classified participants. Even here the mean classification performance is with 75% above chance level.

**Author response image 2. sa2fig2:** Left/right decoding performance for a conjoint classification of PAM/SAM EMG using an individualized decoding scheme in Experiment 2.

**Author response image 3. sa2fig3:** Left/right decoding performance for a conjoint classification of PAM/SAM EMG using an(non-individualized) decoding scheme in Experiment 2 with a leave-one-participant-out cross validation.

The results for the very same “leave-one-participant-out” cross validation is shown in Author response image 4 Experiment 1, i.e., the exogeneous attention setting. Here we used the rms-value of the entire response to generate the learning/testing associations for the support vector machine. We used a training set of 486 observations (subjects^-1^*nstimuli*directions/27*9*2) and an independent test set of 18 observations (nstimuli*directions/9*2). Also here the classification accuracy is far above chance level.

**Author response image 4. sa2fig4:** Left/right decoding performance for a conjoint classification of PAM/AAM/SAM/TAM EMG using an (non-individualized) decoding scheme in Experiment 1 with a leave-one-participant-out cross validation.

4) For Experiment 2, EMG activity is basically averaged across the entire 5 min range. This is a very coarse timeframe and approach, and I can't help but think there would be something more interesting in the data. Is there any way of looking for transient changes in auricular muscle recruitment, and then tying that back to some sort of event during the stream of auditory information? There is the chance of some potentially rich data that really hasn't been mined with the current approach.

The reviewer is posing an excellent question regarding events which trigger particular movements. This is certainly a subject for future research, which we now note in the Discussion section. However, as we focused on sustained spatial listening in Experiment 2, the speech material was not designed for segmentation into events. In fact, we have chosen speech material that has a rather balanced saliency, arousal level, and a homogeneous information density. There was also, as described, the freedom for participants to pick a story and, consequently, the attended material varied across participants. Nevertheless, we describe more carefully in the revised version that Figure 6 and Figure 7 have a time-resolved analysis (resolution of 10s segments) that shows a sustained activation for the course of the experiment.

[Editors’ note: what follows is the authors’ response to the second round of review.]

Reviewer #3:General assessmentThis short manuscript reports that tasks that engage auditory attention either exogenously (Experiment 1) or endogenously (Experiment 2) lead to the recruitment of auricular muscles that subtly change the shape of the pinna, doing so in a spatially-dependent manner. The conclusion is that such recruitment attests to the presence of a vestigial brain circuit. The topic is timely given recent findings linking saccadic eye movements to movements of the eardrum, and a number of other subtle indicators of covert attention driven, for example, by the oculomotor system. The results are intriguing, but more can be done to address other potential confounds, particularly on the oculomotor side.

The authors would like to thank reviewer 3 for the critical feedback and the suggested new analyses to strengthen our argumentation regarding possible confounds between the auriculo- and oculomotor system or other muscular co-activations. The suggested new analyses were really a clever set of ideas and the results are now reported in the new submission (see figure supplements specified below). The authors really appreciate this interest in oculo-auriculomotor interactions. The suggested analysis techniques along with the recently increasing interest in these interactions provide a solid plan for future research in the involved research labs and hopefully for others too. But as discussed already, our study was really designed for the analysis of auricular muscle activation during spatial auditory attention and not the analysis of the interaction between the auditory and visual systems per se. We merely wanted to monitor relationships between eye and ear movements that are currently known to exist: the oculo-auricular phenomenon of Wilson. In this way, we could exclude that the described effect is secondary to the Wilson’s phenomenon (as we understand it today) or to gross neck movements. Therefore, we tried not to over-analyze our data regarding oculo-auriculomotor interactions and drawing possibly too steep conclusions. The authors really appreciate that reviewer 3 (and the editors) left room for this. Thanks to reviewer 3, the limits of our techniques regarding the oculomotor system are now carefully discussed in the manuscript. Our discussion states clearly that exploring these interactions will be an interesting future research path. Unlike the present studies, which included a fixation cross or reading task, it would be better to employ a free-gaze paradigm. This would maximize the chances of observing oculo-auricular co-activation.

Substantive concernsThe authors have extensively revised the manuscript, and established the phenomena both within and across their subject pool. I still have some concerns about other potential confounds that need to be addressed; as the authors say the neural drive to the ear muscles is so weak that the resultant movements are miniscule compared to those generated during broad smiles or wiggling.1) Previously, I had raised concerns about potential confounds from the oculomotor system that orients the line of sight via eye and/or head movements (what the authors term the "visuomotor system" in their response). The authors have added a number of analyses that go some to length to assuage concerns about eye movements. However, grand average measures of EOG across many trials could mask some interactions between eye movements and auricular muscle activity; the data shown in Figure 7 also shows how the variance of the EOG signal decreases after stimulus onset, particularly for stimuli presented at the left-back speaker. More analyses and details are warranted.

As mentioned above, we added all the suggested new analyses. In particular, we extended supplements of Figure 7 and 8, see the specifications below.

1a) For Experiment 1, the auditory stimuli are presented while subjects are "reading a boring essay". Please provide details about how large the eye movement excursions were; from Multimodal Figure supplement 1, it appears that the text spanned about +/-12 deg of horizontal visual angle, but this is just one subject.

We have now included the box-plot analysis for all the subjects, see Figure 7—figure supplement 2.

1b) The authors acknowledge that they can detect "macro" saccades greater than about 1 degree on average, and these should be analyzed in a more systematic manner than relying on average EOG traces, which could wash out effects. The oculomotor literature on microsaccades has a number of ways of presenting spatial and temporal patterns of saccades timed to external events (e.g., see saccadic “rasters” in Figure 4 of Tian, Yoshida and Hafed Front Syst Neurosci 2016), and I think these should be applied to the data from Experiment 1. See for example the work by Ziad Hafed (Figure 4 in Tian, Yoshida and Hafed Front Syst Neurosci 2016). Something similar for the "macrosaccade" data from Experiment 1 could is needed to establish whether or not stimulus onset is altering the patterning of larger saccadic eye movements.

The authors would like to thank reviewer 3 for this excellent suggestion. We included a similar (macrosaccadic) raster plot. This shows a rather uniform distribution of the macrosaccades along the time axis/after the stimulus onset, see Figure 7—figure supplement 3.

1c) For Experiment 2, please provide the "time-resolved" plots for EOG data, similar to what is provided for auricular muscle activity in Figure 6. The EOG data shown in Figure 8 is helpful, but shows data averaged across an entire 5 min segment I believe, which is a very large window.

We agree. A time-resolved plot, similar to the one used for the auricular muscles, is now shown in “Figure 8—figure supplement 1”.

2) In regards to a potential concern about head movements, I agree that the vestibular-auricular reflex is unlikely, given that the head was stabilized on a chin-rest. My previous concern was more about whether the act of spatially deploying auditory attention was related to neck muscle contraction that introduced cross-talk at the auricular muscles (the absence of head movement can't be used to infer the absence of neck muscle contraction in this regard). The work by Cooke and Patuzzi, 2014, doesn't address this concern since they examined sternocleidomastoid activity during ipsilateral head turns (right PAM and right SCM recordings during right head turns). SCM is a contralateral, not ipsilateral head turner, so would be directly recruited by leftward turns, which do not appear to have been studied in the Cooke and Patuzzi setup. The Cooke and Patuzzi paper actually mentions the concerns I have about potential cross-talk from other nearby muscles (see end of the first paragraph of their "subjects and methods"). Muscles on the back of the neck (e.g., splenius capitis or suboccipital muscles; insertion on the occiput, which lies close to the mastoid) are ipsilateral head turners. The activity of the suboccipital muscles in particular has also been related to reflexive visuospatial attention in a Posner type task in head-restrained monkeys (e.g., see Corneil et al., 2008). To be clear, I don't think that the entirety of the results could be "explained" by cross-talk from nearby dorsal neck muscles, but the authours should consider this perspective.

The authors agree, even though our bipolar configuration of electrodes would be rather robust to the possible volume conduction effects mentioned in the Cooke and Patuzzi paper. Also, the fact that far apart auricular muscles (not just PAM, but also AAM and SAM) show these effects is of course promising with respect to a co-activation interpretation. We note in the revised manuscript the possibility that subtle, covert activation of head turning muscles as suggested by the reviewer might be correlated with ocular and auricular orienting. Thanks for mentioning this and for the reference. We certainly did consider this perspective, see the last paragraph before the discussion.

3) A final point about other muscles; Figure 1 shows that the authors recorded EMG activity on zygomaticus major and frontalis, but results are not analyzed. Given the point about how small the movements of interest are compared to those related to smiling, please establish the independence of the auricular muscle recordings from these facial muscles.

Indeed, we had electrodes at these positions, mainly for a different subsequent experiment that analyzed the interaction of these muscles and the auricular ones during positive affective states. The frontalis and zygomaticus data are now reported for Experiment 1 and 2, see Figure 7—figure supplement 6-9 and “Figure 8—figure supplement 2”, respectively.